# The clock in growing hyphae and their synchronization in *Neurospora crassa*
Jia Hwei Cheong [1], Xiao Qiu[2], Yang Liu [1], Emily Krach[3], Yinping Guo [3], Shishir Bhusal[4], Heinz-Bernd Schüttler[4], Jonathan Arnold [3] ✉ & Leidong Mao [5]

Utilizing a microfluidic chip with serpentine channels, we inoculated the chip with an agar plug with *Neurospora crassa* mycelium and successfully captured individual hyphae in channels. For the first time, we report the presence of an autonomous clock in hyphae. Fluorescence of a mCherry reporter gene driven by a *clock-controlled gene-2 promoter* (*ccg-2p*) was measured simultaneously along hyphae every half an hour for at least 6 days. We entrained single hyphae to light over a wide range of day lengths, including 6, 12, 24, and 36 h days. Hyphae tracked in individual serpentine channels were highly synchronized (K = 0.60–0.78). Furthermore, hyphae also displayed temperature compensation properties, where the oscillation period was stable over a physiological range of temperatures from 24 °C to 30 °C ($Q_{10}$ = 1.00–1.10). A Clock Tube Model developed could mimic hyphal growth observed in the serpentine chip and provides a mechanism for the stable banding patterns seen in race tubes at the macroscopic scale and synchronization through molecules riding the growth wave in the device.

One of the central problems in systems biology[1] is understanding how clocks in single cells synchronize—that is, can we explain how the irregular behavior of single-cell clocks gives rise to the highly orchestrated behavior of populations of $10^7$ cells per ml[2,3]. This synchronized behavior of cell populations is observed in plants[4], flies[5], worms, and mammals[6] as well as fungi[7,8]. The answer to this question may depend in part on which life stage we observe the clock to operate. Our initial efforts to address this question have focused on nondividing conidial cells[3]. We initially restricted our research to the conidial stage to simplify the mechanisms at work in cell-to-cell synchronization. We restricted cell growth with specialized minimal medium[2,9] to suppress cell division and the complication of the circadian gating of cell division[10,11]. In this paper, we promoted cell growth to examine other mechanisms of cellular clock synchronization.

As these conidial cells progress to later developmental stages, there is the possibility of other mechanisms for cellular synchronization absent in single conidial cells coming into play, such as cell cycle gating of circadian rhythms[10–12]. Three mechanisms of clock synchronization may be at work in single nondividing conidial cells. Firstly, quorum sensing (QS) acts by secreting signaling molecules into the surroundings at a specific density to synchronize cells[3]. Alternatively, cells could communicate by cell-to-cell contact as in Myxobacteria[13] or possibly the filamentous cyanobacterium, Anabaena[14]. Third, stochastic intracellular noise is vital in generating stable circadian oscillations[15,16]. However, with cells growing as hyphae, there are at least three additional mechanisms for synchronization[2]. A QS signal could be produced and transported solely between nuclei in the same hyphae, or this signaling molecule could be produced and transported into the media. Lastly, the nuclear division could gate the synchronization process[10,11]. Exploring the collective behavior of hyphae requires new kinds of experimental approaches in microfluidics[17,18] as well as new theoretical approaches to the study of cellular clock synchronization in hyphae[19]. Understanding how molecules within a syncytium of nuclei behave in a fungal network is part of a much broader problem of understanding how mRNAs and proteins are shared between nuclei under a balance of advection (i.e., drift within the shared cytoplasm), diffusion of these molecules[20] and other active transport processes[21].

Understanding how the clock arises in filaments rests on physical processes of passive displacement[21] of nuclei known as diffusion and migration (i.e., the active transport) of nuclei by molecular motors[21]. There are two views of this underlying physical process. The first view is that nuclei generate throughout the syncytium and gradually move to the tip through a mixing process. Another hypothesis is that nuclei divide subapically adjacent to the nuclear exclusion zone in a region we call the *advection zone*, thereby populating the growing hypha by riding the growth wave. The challenge in distinguishing these hypotheses is detecting nuclear movement away from the growth tip. Hence, the serpentine microfluidics device capable of capturing several filaments should address these questions.

The goal here is to explore four questions: (1) Can a Clock Tube Model be developed and specified for a single or few hyphae observed by

[1]Chemistry Department, University of Georgia, Athens, GA 30602, USA. [2]Institute of Bioinformatics, University of Georgia, Athens, GA 30602, USA. [3]Genetics Department, University of Georgia, Athens, GA 30602, USA. [4]Department of Physics and Astronomy, University of Georgia, Athens, GA 30602, USA. [5]School of Electrical and Computer Engineering, College of Engineering, University of Georgia, Athens, GA 30602, USA. ✉e-mail: arnold@uga.edu

microfluidics; (2) how do hyphae and their nuclei behave over at least a 6-day time scale; (3) can we identify the advection zone in hyphae; and (4) does the Clock Tube Model developed here behave like real hyphae as well as provide a mechanism for spatially stable banding patterns and phase synchronization? The beginning point of this work is merging growth models for filamentous fungi[22,23] with existing deterministic clock models on conidia with experimental support[24].

Here, we developed a microfluidic device with multiple serpentine channels, each of 32 mm, to address these four questions. Previous work has only identified hyphal behavior over time scales of less than a day[17,18]. The present study aims to identify the presence of clocks in single hyphae with the setup shown in Fig. 1, with measurements taken over at least 6 days. The best-studied clock-controlled gene is *clock-controlled gene-2* (*ccg-2*)[25]. MFNC9 strain possesses part of the promoter of *ccg-2* attached to a mCherry reporter without the upstream region providing developmental control of *ccg-2*[3,25]. The remaining upstream part provides White Collar Complex

(WCC) control by the clock. Cells engineered for fluorescence[26] were inoculated into a microfluidic device and placed on a Zeiss microscope in a temperature-controlled incubator. Media was pumped through the microfluidic device to support filament growth and imaged every 30 min over at least 6 days. Our results help us understand how they behave and apply them to validate a clock model for growing hyphae.

## Results

### Serpentine microfluidic device design provides a growth environment for filaments

The microfluidic device was designed to provide growing cells with sufficient media for growth and recording their fluorescence over time using fluorescent strain MFNC9[26]. The diameter of *N. crassa* conidia is known to be around 3–8 μm[2], while the hyphae have a diameter of around 8–15 μm. The height of 10 μm was designed for the serpentine growth channels to ensure that their height is slightly larger than the *N. crassa* hyphal diameter, hence ensuring that media could constantly flow through the serpentine channels. We verified this by simulating the flow profile with COMSOL Multiphysics software (Supplementary Fig. S1). The width of the chip was maintained at 16 μm to ensure the possibility of media transported along the channels via diffusion to support a growing filament. Reducing channel width further, however, would adversely affect the flow of media through the channel, affecting filament growth differentially along the channel.

To observe hyphal growth for at least 6 days, we fabricated serpentine chips of an additional length of 63 mm. We loaded an agar plug inoculated with conidia into the cell inlet and sealed it with a modified flexible tubing of the same diameter. Hyphae grew from the agar plug and elongated towards the serpentine channels due to the presence of nutrients supplied from the medium channel at the other end of each serpentine channel. Initial amplitude variation in Fig. 2a, b in the first 2 days is consistent with race tubes and single filaments entering a stable limit cycle. We observed clear oscillations for serpentine chips, as seen in Fig. 2b. Hyphae were observed and tracked in both chips, and hence in the future, depending on the studies that need to be conducted, either chip can be utilized.

We constantly supplied media into the chip to ensure hyphae have access to nutrients. We examined the influence of varying glucose concentrations (0.1%, 0.5%, 0.70%, 1.5% glucose) in the serpentine chip. The goal was to ensure cells were provided with an adequate amount of nutrients to grow as well as to oscillate. Interestingly, when a glucose concentration of 0.5–1.5% was infused into the chip, hyphae only grew a quarter length of the serpentine channel and ceased to grow the entire length. An explanation for this phenomenon is that when we increase the glucose concentration, cells efficiently take in more of the nutrients provided; this ultimately increases the thickness of filaments in the channels, resulting in a considerable pressure buildup in the chip, as seen in Supplementary Fig. S2. As a result, media would have difficulty diffusing or flowing through compared to when we use a lower glucose concentration. Hence, considering that the elongation of hyphae at a higher glucose concentration was inhibitory to hyphal extension, for further experiments, 0.1% of glucose media was supplied into the chip. Hyphae were able to grow the full length of the channel, and time-lapse experiments of at least 6–10 days were successfully executed (Fig. 1).

### Clock tube model in a growing hypha

The motivation of the model is threefold (see Model Supplement for details). The first and foremost reason for the model is to provide a conceptual framework for examining the clock and other biological processes in the dominant life stage of the organism, the growing filament. Most previous work on the clock in *N. crassa* at the molecular level focuses on liquid conidial cultures. Consideration of filaments forces a consideration of both the temporal and spatial aspects of the clock, as one might see in race tubes. The model must have both of these aspects for validation of the model. When considering the growth of a filament, physical processes, such as diffusion, advection, and Taylor dispersion, come into play, and it is an open question how these physical processes impact the clock. A hyphal clock model provides a framework to explore their connection to the clock.

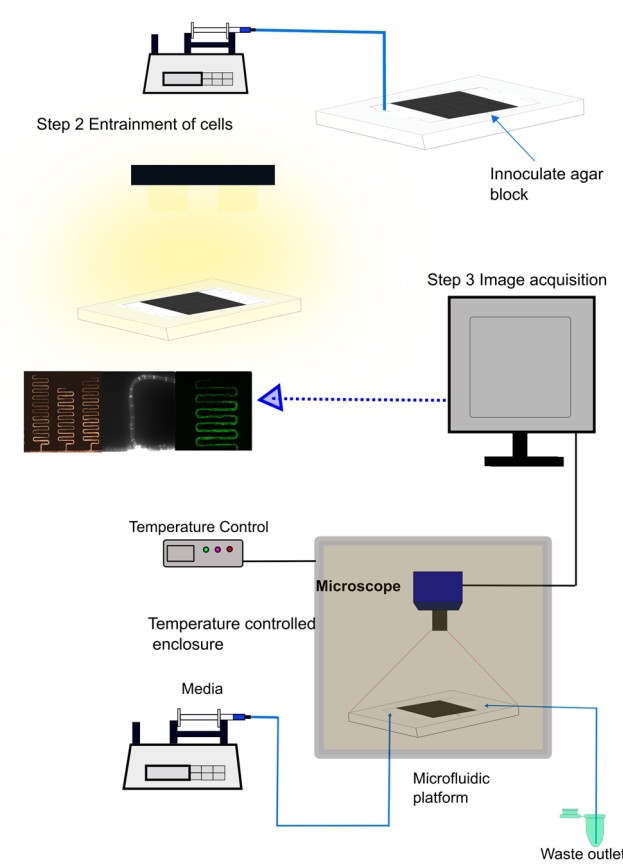

**Step 1 Cell innoculation**

**Step 2 Entrainment of cells**

Innoculate agar block

**Step 3 Image acquisition**

**Temperature Control**

**Microscope**

**Temperature controlled enclosure**

**Media**

**Microfluidic platform**

Waste outlet

**Fig. 1 | Schematic representation of the experimental setup.** In step 1 of the protocol, media was loaded into the chip with a syringe pump continuously till the whole chip was filled. An *N. crassa* agar block with fluorescent strain MFNC9[26] was then loaded into the cell inlet port located at the lower central position, as shown in Fig. 13. To prevent the agar block from drying out, a 3D printed plug was used to seal the cell inlet port. In Step 2, the chip containing the agar block was placed under a light source with at least 2 h of light at 5370 lux. After exposure to light, the microfluidic platform was placed onto the microscope stage located inside a temperature-controlled enclosure. Throughout each experiment, media was constantly supplied to the microfluidic platform with a syringe pump. In step 3, filament growth of *N. crassa* cells was viewed (micrograph on the left) by time-lapse fluorescence imaging with a time interval of 30 min. For specific temperature compensation experiments, we varied the temperature with the temperature control instrument while ensuring the device was maintained in the dark. Meanwhile, an LED light source was placed into the enclosure and used for light entrainment experiments while maintaining a constant temperature. Time series of fluorescence measurements over at least four days were obtained through the images collected.

**Fig. 2 | Detrended trajectories for the luciferase strain *frq-luc-I* in race tubes[7] and MFNC9 strain in a microfluidic device.** Hyphae of fluorescence strain (MFNC9) tracked with the FT method described below (**b**) yielded a circadian rhythm comparable to that of the luciferase strain *frq-luc-I* in race tubes[7] (**a**), the average signal-to-noise ratio (S/N) as described in Materials and Methods, for the fluorescence versus luminescence strain was 12.09 and 33.39 for the latter. The *frq-luc-I* graph was generated from supplementary movies[7] (see Materials and methods). Both images were log-detrended with a moving average described earlier[2]. The average periods of bands and hyphae are **a** 20.78 h; **b** 19.20 h.

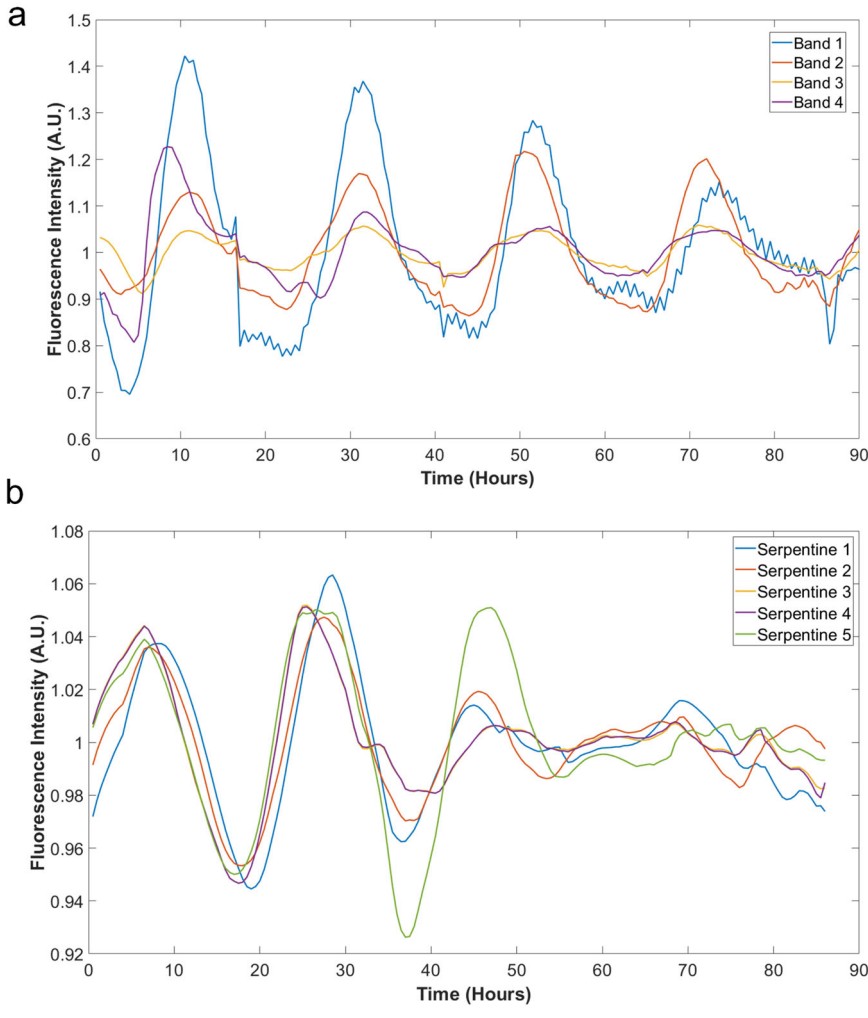

The final motivation for a hyphal clock model is that it provides a conceptual framework for what to measure about the clock in a growing filament. The physical processes are coupled to more standard genetic networks describing conidial cultures microscopically in microfluidics devices, and the importance of physical processes must be captured by measuring diffusion rates, advection rates, and Taylor dispersion to understand process effects.

Once these physical processes are known and measured, this gives rise to hypotheses about how they impact the clock. For example, where is the clock functional? Is it functional in an advection zone of a growing filament, and how does it behave outside of this zone? The concept of an advection zone comes from the modeling framework coupled with measurements on advection. Is advection and diffusion sufficient to explain long-distance synchronization in a microfluidics device? This kind of question can only be answered in the conceptual framework of the hyphal clock model, which includes advection and diffusion as processes. Are there new ways to conceptualize how synchronization of clock filaments arises over long distances? Again, the dynamics of an advancing wave of cytoplasm within a growing filament may provide an answer about long-distance synchronization that could not be envisioned without the model. These are the reasons that a model must be considered for a growing filament.

We consider a spatiotemporal clock model in one spatial and one-time dimension, with position and time coordinates $x$ and $t$. The model describes either a single growing hyphal cell, modeled as a long, thin cylinder as shown in Fig. 3, or a bundle of such hyphae, located in close proximity, co-oriented in space, and evolving in approximate synchrony in time. Each hypha comprises a large population of intra-cellular nuclei, carrying the essential clock genes and modeled by $x$- and $t$-dependent nuclear density distributions. The hyphal cell also comprises all genetic and molecular species participating in the clock oscillation and signaling mechanism, as shown in Fig. 4, modeled by respective $x$- and $t$-dependent molecular concentration profiles and gene state density distributions.

The model is designed to allow for one-dimensional spatial concentration and density patterns to evolve as a function of time along the cylindrical $x$-axis of the hyphal filament. The crucial element of the model to generate such spatial patterns is the coupling of the temporal clock oscillations to the advection of intra-cellular molecular and nuclear (gene) clock components along the hyphal filament. This intra-cellular advection is driven by the inflow of a liquid extra-cellular medium and the resulting flow of the intra-cellular liquid towards the hyphal growth tip, modeled by a flow velocity profile $v(x, t)$.

An additional crucial feature of the model is the inclusion of continual mitotic nuclear division inside the hyphal cell. These nuclear division processes are crucial for the clock-generating molecular and genetic species to be created and populate the continually expanding hyphal cell volume that is created by the growth of the hyphal cell itself.

The intra-cellular advection and nuclear division is coupled to the "standard" clock oscillation mechanism, described in earlier work[27]. As illustrated in Fig. 4, the clock oscillations are generated through the negative feedback loop comprising the gene activator protein complex *WCC*, its antagonist protein *FRQ*, and their respective gene and mRNA precursors[28,29]. The resulting spatio-temporal model is a system of coupled partial differential equations (PDEs)[22,23]. It is described in detail in the Model Supplement, including the complete model parameter set and the numerical

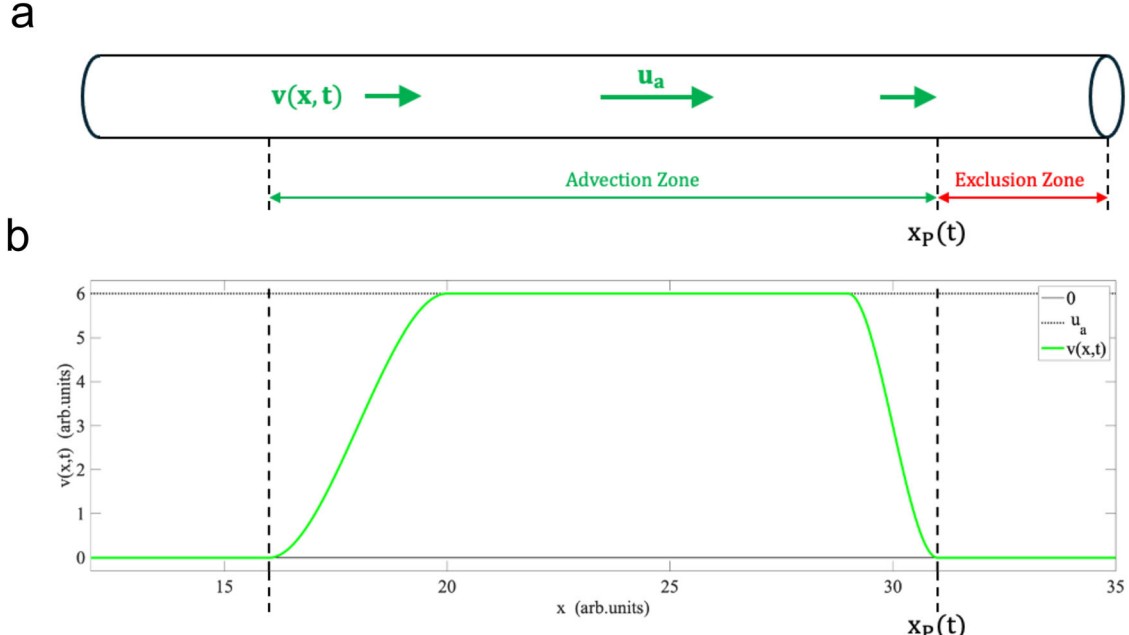

**Fig. 3 | Graphical representation of the clock tube model. a** A growing hypha idealized as a cylinder is shown. **b** The intra-cellular flow velocity profile, $v(x, t)$, along the hypha is shown schematically. The influx of nutrients and liquid media drives the intra-cellular flow toward the hyphal growth tip. Vertical dashed lines mark the left and right boundary of the advection zone, where the point of reference $x_P(t)$ denotes the moving growth front. The maximal velocity in the advection zone is denoted by $u_a$. The velocity profile approaches zero at the beginning of the exclusion zone and sufficiently far back from the tip. The velocity profile plotted here is schematic and does not reflect quantitatively the actual $v(x, t)$ detailed in the Model Supplement and used as input in the model solution shown in Fig. 11.

PDE solution method which we have employed to generate the model results shown in Fig. 11 below. The implementation of the model is also described in a manual for the MATLAB code we have used to compute the PDE solution.

The flow velocity profile, $v(x, t)$, is displayed schematically in Fig. 3, to illustrate its main qualitative features. The exclusion zone is the segment about 12 μm before the hyphal growth tip. This section contains the Spitzenkörper and the subapical region to the front edge of observed nuclear movement[21]. To a first approximation, the flow velocity profile $v(x, t)$ at position $x$ in the hyphae at time $t$ is considered to have a fixed shape, which decreases to zero sufficiently far back from the tip. This segment of nonzero velocity, $v(x, t) > 0$, is referred to as the advection zone. The maximal velocity in the advection zone, denoted by $u_a$, is assumed to be equal to the tip growth velocity. As a function of time, $t$, the entire flow velocity profile moves at constant velocity in the $(+x)$-directing thereby tracking the position of the moving growth tip, as required by the basic physics of incompressible fluid flow into an expanding volume.

**Variation of hyphal number growing along serpentine channels**
The strain (N2281-3) with a fluorescent reporter on the H1 Histone was used to examine nuclei within hyphae[30]. This method allowed us to observe nuclei moving along hyphae, which provided us with a better observation on the number of hyphae present in serpentine channels. From the Supplementary Movie S1, we observed varying velocities in the movement of nuclei in each of the hyphae. A total of 1–4 hyphae were growing along each serpentine channel. To further verify these results, we stained hyphae grown in channels with calcofluor white. It is known that calcofluor white is a nonspecific fluorochrome that will bind with chitin and cellulose in the cell walls. Hyphal cell walls were stained by injecting 0.01% calcofluor white and 0.1% KOH solution into the serpentine chip and incubated for 10 min at room temperature (RT) in the dark. Images were taken under an inverted microscope with a filter set 49 (Zeiss) (see Materials and methods). The stain effectively differentiated the number of hyphae present in the serpentine channels as they grew. In all images, we observed intense staining of cell walls and septae in individual hyphae. We could clearly differentiate the

hyphae's boundaries with the stain (Fig. 5). We also conducted a control experiment with a wild type (WT) strain to confirm our findings. The control experiment confirmed that the mCherry fluorophore did not interfere with the Calcofluor white staining in the MFNC9 strain. We concluded that 1–6 hyphae were growing into the serpentine channels.

**Optimal strategies for tracking hyphae were developed**
This section investigates the performances of three hyphal tracking methods to identify the best tracking method. One method—hyphal tip tracking (HTT) was to measure the extension of the hyphal tip exclusively as it was growing. We tracked three hyphal tips individually as they grew along the serpentine channel (Supplementary Fig. S3) for a period of 30 h. The fluorescence intensity results were very noisy—if oscillations were present, they were relatively weak and not very noticeable (Supplementary Fig. S3). Next, we examined cropped segments across the tiled image horizontally. The segments were identified by the number of U-turns (1, 2, 3, or 4 U-turns) across the serpentine channels in a field of view as presented in Fig. 6. We can observe a similar pattern for at least 2 U-turns—4 U-turns; however, the periodogram on 1 U-turn failed to capture the features seen in the other three periodograms with multiple U-turns. Hence, we deduced that 2 U-turns—segment tracking (ST) will be sufficient when we carry out the data analysis process. Another reason to use 2 U-turns vs. 3 U-turns or more was that we would not lose any spatial-temporal information. Trajectories of varied segments along a serpentine channel also generally aligned with each other (Supplementary Fig. S4) for 98 h.

The last method—filament tracker (FT) was used to track individual hyphae as they grew across the entire channel length (Supplementary Fig. S5). We observed clear oscillations in individual serpentine channels, which aligned with each other. These results imply that the hyphae are synchronized across long distances. Ultimately both tracking methods, including ST and FT methods, are recommended.

**The behavior of hyphae in a microfluidic device**
We sought to compare the behavior of hyphae for the fluorescent strain (MFNC9[26]) in the serpentine microfluidics device with a luminescent strain[7]

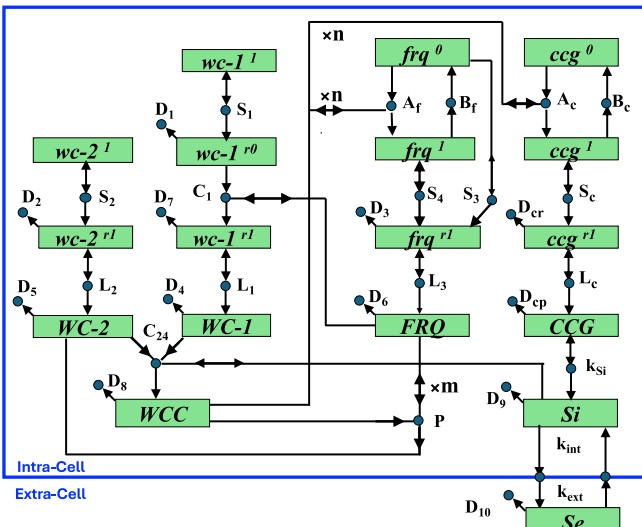

**Fig. 4 | Genetic network for the clock in *N. crassa*.** Rectangles are species, i.e., genes, mRNAs, proteins, or signaling species. Circles denote reactions. An arrow from species to reaction indicates a reactant species. An arrow to species from reaction indicates a product species. A species connected to a reaction by double arrow is catalytic. Reactions with no products are degradation reactions. Proteins are in all capitals. Superscripts "1" and "0" indicate gene states with high and, respectively, with low or zero transcriptional activity, due to binding and, respectively, unbinding of *WCC* at the *frq* or *ccg* activator binding site. Superscripts "r1" and "r0" indicate translationally active and inactive mRNA, respectively. The labels of reactions also serve as rate coefficients for the cognate reaction. Reaction labels starting with letters *A, B, S, L, D,* and *C* indicate, respectively, gene activation (by WCC binding), gene deactivation (by WCC unbinding), transcription, translation, degradation, and conversion or complex formation. Reaction *P* is the FRQ-catalyzed degradation of *WCC*. Reaction $k_{Si}$ is the *CCG*-catalyzed production of intra-cellular signaling species; $k_{int}$ and $k_{ext}$ are the rate coefficients for intra- and extra-cellular signaling species, $S_i$ and $S_e$, respectively, to diffuse between intra- and extra-cellular space through the cell membrane. Species enclosed by the big box reside in the intracellular space of a single growing hypha or a bundle of such hyphae. Quorum sensing is enabled by the exchange of the signaling species between intra-cellular and extra-cellular space. The figure is taken from earlier work[27].

grown in race tubes over 5 days (Fig. 2). Previous work by Dunlap and others have shown that circadian oscillations are present by tracking its luminescent bands in race tubes (Supplementary Fig. S6).

Here, we showed that hyphae behaved similarly and had a similar period, indicating the presence of circadian oscillations in both scenarios (Fig. 2). In both cases, the oscillations are synchronized. The average signal-to-noise ratio (S/N) of race tubes is about 3X that of hyphae growing in serpentine channels. We selected 8 random hyphae in individual serpentine channels and calculated the S/N to obtain some idea of variation within a microfluidic device. We show the results in Supplementary Table S1. The key points are that there is a variation of the S/N but the period for MFNC9 (22 h) is similar to that in race tubes[26] (21 h). S/N of hyphae overlapped with that of race tubes with a luminescent strain (Fig. 2).

## Spatial-temporal dynamics on serpentine chip

Time-lapse images obtained during confined growth in the serpentine channels were used to build a spatial-temporal diagram using 26 boxes, such as the one displayed in Fig. 7a. This graphical representation allowed us to visualize the variation of fluorescence intensities spatially across the channel. Interestingly, on such a diagram, as we look across the time series for each individual position, there is no significant shift in the pattern of the fluorescence intensities. We do not observe oscillations or huge changes at positions that present a peak. These peaks seem to occur at most of the curved segments in the channel. To further explore the growth dynamics observed, we examined a particular horizontal section of the serpentine

channel. We divided the image into 12 boxes. As seen in Fig. 7b a higher fluorescence intensity is recorded at both ends of the U-turn compared to the straight channels.

## Synchronization of hyphae in serpentine channels

Our previous work showed that synchronization exists in a "big chamber device" and microwell devices[27] while cells were in their conidial form. We sought to explore further whether we can observe synchronization through growing hyphae in their serpentine channels along the whole device. We examined the synchronization between channels as shown in Supplementary Fig. S7. We used a synchronization measure known as the Kuramoto order parameter $(K)$[31] to compare the synchronization between different segments. The equation to compute K is defined as:

$$K = \left\langle \left| n^{-1} \sum_{j=1}^{n} \exp\left(iM_j\right) - \left\langle n^{-1} \sum_{j=1}^{n} \exp\left(iM_j\right) \right\rangle \right| \right\rangle$$

where $K$ represents the phase coherence, and $M_j$ is the phase of the jth serpentine segment with hyphae. Quantity $n$ is the number of hyphal segments being compared ($n = 2$ for two hyphae segments in neighboring serpentine channels). The order parameter $K$ would be 1 if both channels were equally phase synchronized, while it would be 0 in incoherent states. The $K$ value between any channel segments was at least 0.7526. The high degree of phase synchronization across channels indicates that hyphae were all synchronized, although they were in separate serpentine channels at long distances, one of the intriguing aspects of filament synchronization. Besides that, the low $K$ value ($K = 0.0322 \pm 0.0007$) calculated on isolated 1-cell droplets[32] was almost near zero, as expected. At the other extreme is a "big chamber device", in which cells were tightly packed into an artificial tissue. In this situation, $K$ was larger than 0.91 between different fields of view in a transect across the artificial tissue[27]. Furthermore, we also looked at the $K$ values for randomly generated white noise oscillations and obtained a result of 0.531 (described in Materials and Methods). The synchronization of filaments in different channels is intermediate between that in isolated 1-cell droplets and a tightly packed artificial tissue in the "big chamber device".

## Hyphae display clock-like properties

With the serpentine microfluidic platform developed and exhibiting circadian rhythms (Fig. 2), we examined two further defining properties of a biological clock, light entrainment, and temperature compensation. We conducted experiments over a range of 24–30 °C, as seen in Fig. 8a. We observed slight variation in the period of the clock over a physiological range, i.e., there is temperature compensation. If temperature compensation were present at the single cell level, the period length of tracked hyphae would not change over the temperature range. This was quantified by the $Q_{10}$ value measure as seen in Equation:

$$Q_{10} = \left(\frac{P_1}{P_2}\right)^{\frac{10}{T_1 - T_2}}$$

where the reference temperature is denoted as ($T_2$) at 30 °C and the periods, $P_1$ and $P_2$, the periods are at temperatures $T_1$ and $T_2$, respectively. The $Q_{10}$ values obtained ranged from 1.0 to 1.1 (Supplementary Table S2), demonstrating small fluctuations of the period with temperature. The minor differences in the $Q_{10}$ value can be attributed to subtle differences in culture conditions[33]. Besides that, the $Q_{10}$ values also fell within the compensated range (<1.3) reported previously[33]. Hence, we revealed that single hyphae demonstrated temperature compensation characteristics.

We want to explore the mechanisms of temperature compensation from data in Fig. 8b. Previous work[3] has shown that there is evidence of amplitude-period coupling that is consistent with presented clock models exhibiting temperature compensation properties[34]. We examined the coupling of the relative amplitude to the relative period in Fig. 8b in over 40 single hyphae. The highly significant positive slope ($p < 0.0001$) of relative

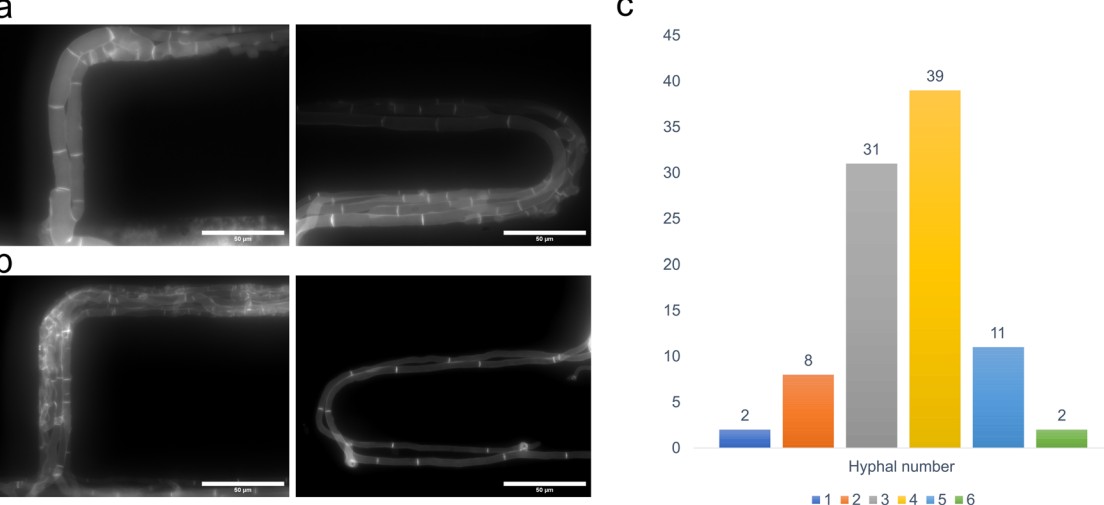

**Fig. 5 | Fluorescent images of hyphae stained with calcofluor white demarcate 1–4 hyphae in a channel. a** Calcofluor white staining on MFNC9 shows two hyphae growing into the opening of the serpentine channels, while the image on the right shows multiple hyphae along the channel's curve. **b** Calcofluor white staining on a wild type (WT strain) shows similar results, with multiple hyphae growing into the serpentine channels. **c** Bar chart representing hyphal number in each serpentine channel (96 serpentine channels) inlet across the serpentine device.

amplitude squared on relative period further strengthens the prediction that coupling is present, as seen in the existing three families of clock models[34,35].

The following attribute of a biological clock is that cells can be light-entrained. We carried out time-lapse experiments of *Neurospora crassa* hyphal growth using a developed upright microscope setup equipped with an LED light source for light entrainment (see Materials and methods). In order to investigate the intrinsic behavior of the clock, we entrained the hyphae to a Light:Dark (L/D) of 3:3 h L/D cycle, 6:6 h L/D cycle, 12:12 h L/D cycle, and an 18:18 h L/D cycle under constant conditions (LED with 5370 lux) and temperature (30 °C). This method allowed us to track growing hyphae in individual serpentine channels while carrying out the entrainment process. As seen in Fig. 9, the trajectories were seen to synchronize to the L/D cycle of the light source for the specified day lengths. One method to estimate the periodogram for fluorescence series is the periodogram or power spectrums[2]. The periodograms for hyphae tracked the period based on the varying day length (6, 12, 24, or 36 h).

### Specifying the velocity profile, drift, and diffusion velocities of the hyphal clock model

Nuclear drift velocity (due to advection), and diffusion velocity are features of the hyphal clock model introduced in the modeling section. These features to be measured are motivated by the model. To study the flow of nuclei in hyphae of *N. crassa* and to identify features of the model for a hyphal clock, we examined nuclear drift velocity (advection) profiles with a histone hH1-GFP reporter over time-lapse microscopy (Fig. 10). The microfluidic setup we introduced to observe nuclei movement measurement differs from previous studies on agar plates[21,36]. Here we isolate a few filaments in individual serpentine channels as they grow toward the food source. First, we focused on observing the drift velocity on nuclei near the hyphal tip. The average nuclear drift velocity in the x-direction calculated was 2.316 μm/ min. In contrast, drift velocity in the y-direction was −0.0014 μm/s, which was almost close to 0 and not significantly different from zero. The diffusion coefficient calculated was half of the slope, 12.97 μm²/min, while in the y-direction, it was half of 3.57 μm²/min.

It has been suggested that the nuclear drift velocity profile along hyphae is expected to diminish drastically or gradually go to zero (Fig. 3) due to the aging of hyphae caused by mechanical stress[37] and the accumulation of intrinsically disordered proteins[38]. As a result, hyphae become vacuolated, septated, or occluded, hence slowing the movement of nuclei[39]. To address whether we can observe a similar outcome of a decrease in the velocity profile, we measured the drift velocity at subsequent segments at the subapical region further away from the hyphal tip. Five serpentine channels were randomly selected to track nuclei movement. We could observe a significant decrease in velocity after a distance of 672 μm (Supplementary Fig. S8), known as the advection zone for all serpentine channels observed. This concept arose directly from the consideration of the hyphal clock model. We performed a Two-Way Analysis of Variance (ANOVA) to analyze the effect of the serpentine channels and distance from the tip on the velocity. The ANOVA test revealed there was a significant difference between velocity profiles at the tip (Group 1) and away from the tip (Group 2) ($p < 0.01$), as seen in Supplementary Table S3. No significant difference was found in the effect of serpentine channels on the velocity ($p = 0.0472$). Additionally, a Kruskal–Wallis test also confirmed the statistical difference ($H(1) = 13.59$, $p = 0.002$) observed between the two groups as defined above (Supplementary Table S4). Hence, this supports our finding of an advection zone estimated to be 672 μm.

We further explored the growth velocity of the hyphal tip. An interesting finding was the average nuclei drift velocity was slower than the growth velocity of the hyphal tip in Fig. 10b when compared to what was previously reported[21]. If nuclear division occurs along the filament, motor transport can make up for the difference in velocity. In the advection zone, the motion of nuclei is in one direction predominantly, with some nuclei moving away from the food source and tip in a retrograde fashion[21] (Supplementary Movie S1). This is consistent with there being both advection for bulk anterograde displacement and diffusion for the bidirectional displacement of the nuclei (Fig. 10).

### The behavior of the hyphal clock model

The spatial-temporal evolution of the *N. crassa* clock system undergoing hyphal growth shown in Fig. 11 uses parameters identified by ensemble methods applied to populations of cells[35] as well as single cells[27], and these model parameters used here are tabulated in the model supplement as well as the supplementary materials for this paper. Solving these partial differential equations is described in the Materials and Methods. The velocity experiments in the previous section can specify new parameters in the model[27] (Fig. 10). As the nuclei divided, there was an initial growth in the density of nuclei until a steady state in nuclear density is achieved along the serpentine channel or race tube (Fig. 11a). The rate of production of nuclei had been established to be once every 100 m on sucrose[40]. We established the velocity profile near the beginning of the growth experiment (Fig. 11b) and used the experimental hyphal growth tip velocity (6.4787 μm/min) in the model. It is

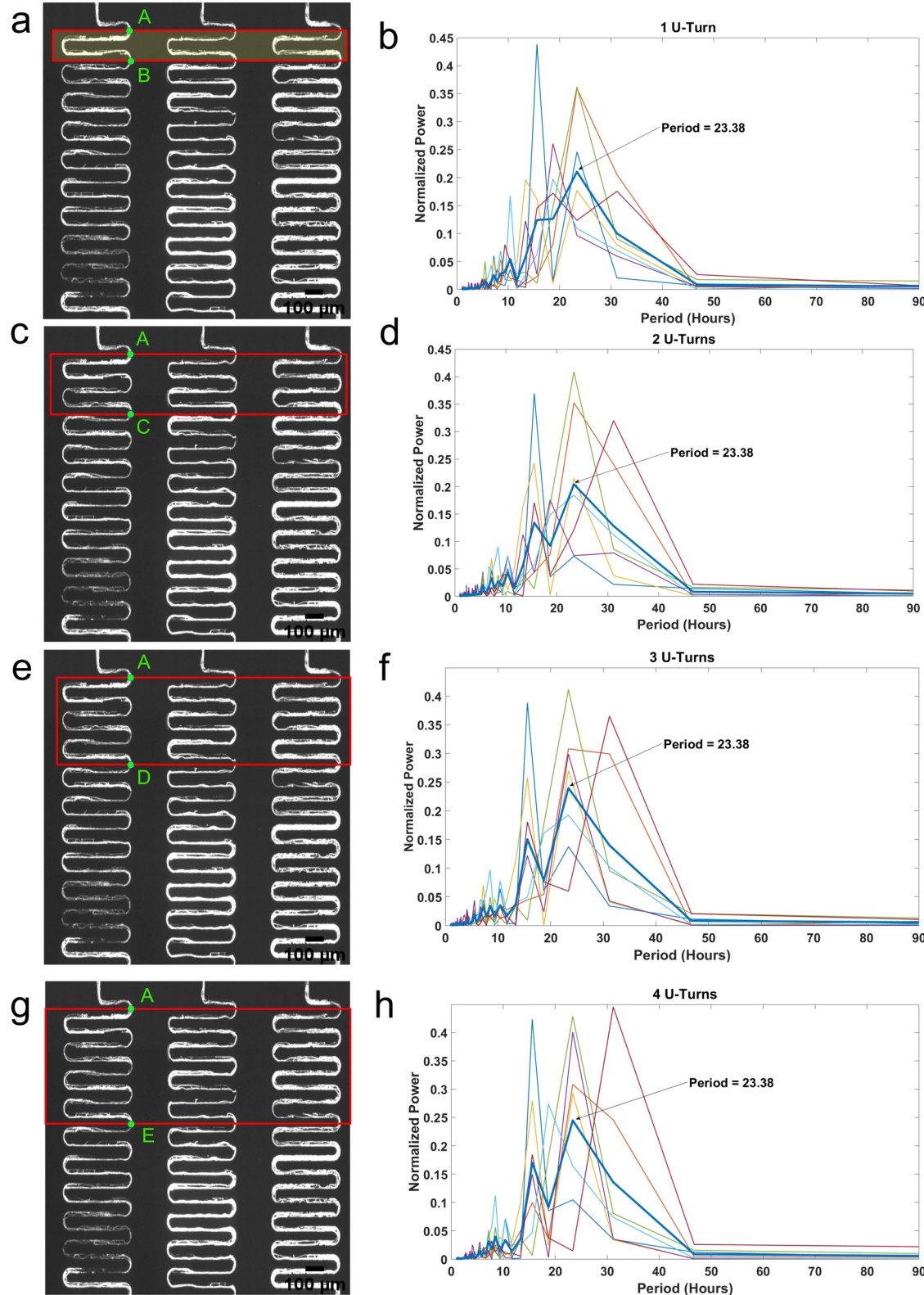

**Fig. 6 | Diagram describing the U-turn hyphal tracking method, developed with CellProfiler. a** Fluorescent image of MFNC9 hyphae growing in serpentine channels. One U-turn is a segment from point A to point B for each serpentine channel. The image is cropped into a red rectangle as depicted before running through the pipeline on Cell Profiler. (**b, d, f, h**) Periodograms of individually tracked hyphae over a period of 4 days in a 32 mm serpentine device. The bold blue line represents the average normalized periodogram, and the finer colored lines represent the periodograms of individual hyphae. **c** Fluorescence image of hyphae growing in the serpentine channels. For each serpentine channel, two U-turns are defined as point A to point C. The length of two U-turns of the serpentine channel segment is 1312 μm. **e** Fluorescence image of hyphae growing in the serpentine channels. For each serpentine channel, three U-turns are defined as point A to point D. **g** Fluorescence image of hyphae growing in the serpentine channels. For each serpentine channel, four U-turns are defined as point A to point E. Scale bar:100 μm.

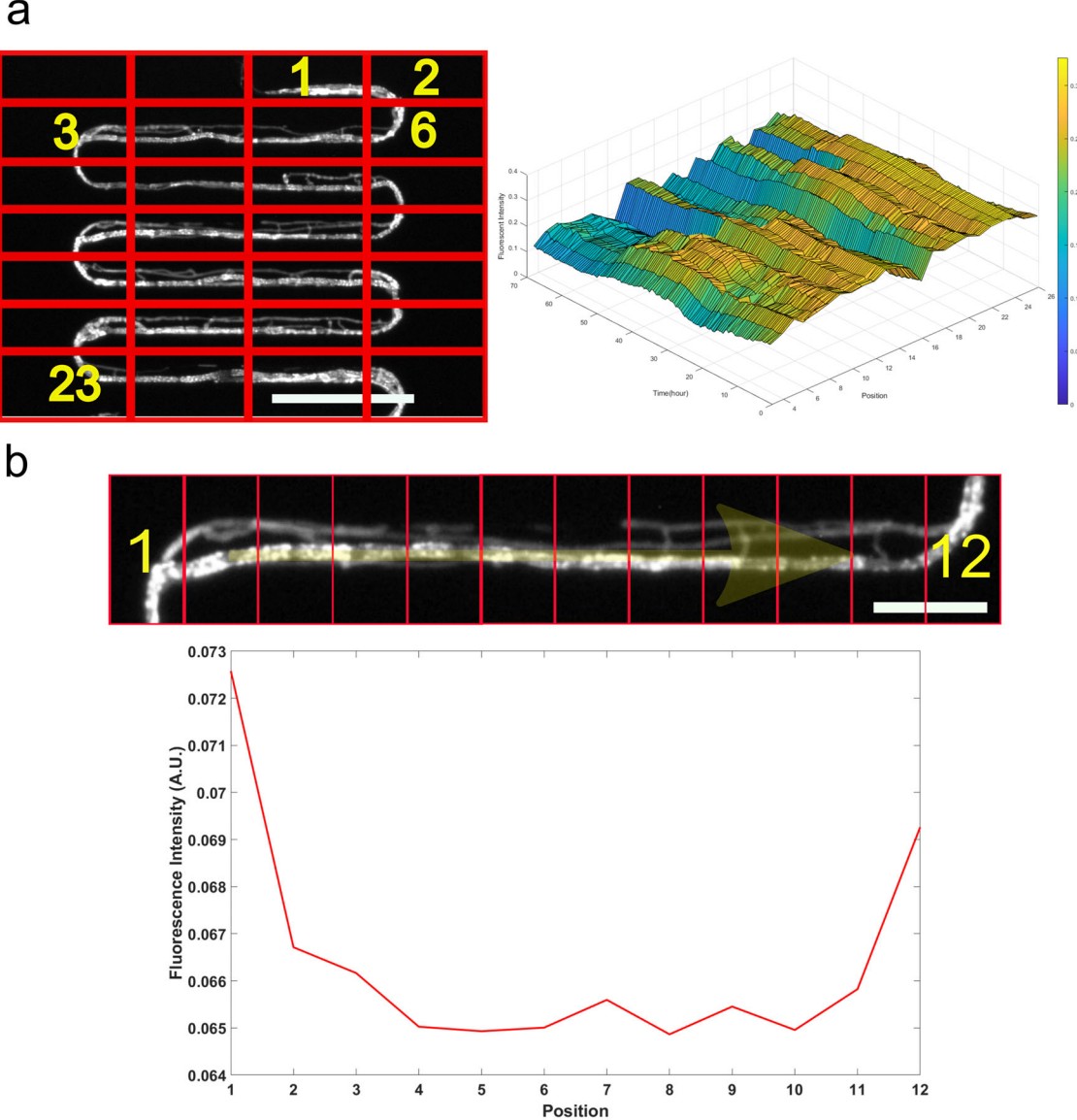

**Fig. 7 | The fluorescence intensity pattern of MFNC9 hyphae varies along position (i.e., segment number) in the serpentine channel. a** Top left: Fluorescence image of hypha divided into 7 × 4 boxes. As shown in the scheme, we labeled 26 positions from left to right, which are in the sequence from top to bottom. Top right: Spatial-temporal diagram of fluorescence intensities over time at different positions. Fluorescence signal intensity is expressed as a gradation, as shown on the right side of the figure; blue indicates a low intensity, while yellow indicates a high intensity. These 26 numbered positions extend over 1968 μm. Scale bar: 150 μm. **b** Top: We labeled 12 positions from left to right on a horizontal segment of the serpentine channel. Bottom: Fluorescence intensities correspond to the labeled positions of hyphae at a particular time point. Scale bar: 75 μm.

near zero at the beginning of the race tube or channel and rises to a constant value in the advection zone as new material is added to growing hyphae. In the model, nuclei are being deposited within the interval of the velocity profile to ensure boundary conditions are obeyed (Fig. 11a). Food density is also constant throughout the whole process and corresponds to the experimental results as media continuously flowed through the device (Fig. 11c). Meanwhile, we can specify the food and velocity profile as a product of step functions in the model to give more flexibility in initial conditions.

Previous studies were done on the measurement of nuclei movement during the growth of hyphae; however, the microfluidic device would introduce additional restrictions for the growth or branching. Therefore, additional data needs to be collected for the hyphal clock model.

The nuclear density, velocity profile, and food distribution are stable over time under the model (Fig. 11) using models fitted to populations of oscillators[35]. Similar results are obtained when we use parameters from single cells in the hyphal clock model[27].

The other aspect of the model is the clock in the hyphae described by a genetic network identified in previous work at the macroscopic level of many cells[35] (Fig. 11e–h). The genes are being swept along in each hypha with the nuclei. They produce mRNAs and proteins along the race tube or serpentine device. Each of these molecular species has a velocity, position, and concentration in the growing hyphae as shown. The model shows the characteristic 21 h period in the FRQ, WCC, and CCG proteins expected in the device. The bands in all three proteins visible in the model have a characteristic period of 21 h, similar to that in the serpentine device (Supplementary Table S1).

There were certain similarities between the observed spatial-temporal profile (Fig. 11d) in the serpentine device compared to the hyphal clock model (Fig. 11g). The velocity profile behaved similarly in the strong diagonal determined by the tip velocity. There was periodic spatial structure visible in both. Lastly, there were circadian rhythms in the t-direction of the correct period (~21 h). The most striking result was the stable "banding

**Fig. 8 | Single MFNC9 hyphae display temperature compensation properties.** The ST method was utilized. **a** The average normalized periodogram at five varying temperatures over the physiological range of *Neurospora crassa* is shown. Each average normalized periodogram at each temperature is based on over 5 (24 °C), 12 (24 °C), 5 (27 °C), 12 (29 °C) and 6 (30 °C) hyphae, respectively. **b** The coupling of the relative amplitude squared plotted vs. relative period. The x-axis is the relative period, calculated by the period of each tracked hypha at the observed temperature and dividing it by the average period at 30 °C. The *y*-axis would be the relative amplitude squared. We calculated this by obtaining the maximum amplitude squared of each tracked hypha and dividing it by the average amplitude squared of the reference temperature (30 °C). Single hyphae are color-coded depending on their temperatures. The correlation ($r$) of amplitude squared and period was $r = 0.2696$ (Fishers $z = 0.2765$, $P < 0.0001$). Spearman rank correlation ($r_s$) was $r_s = 0.3886$ ($P = 0.0132$). The FT method was used.

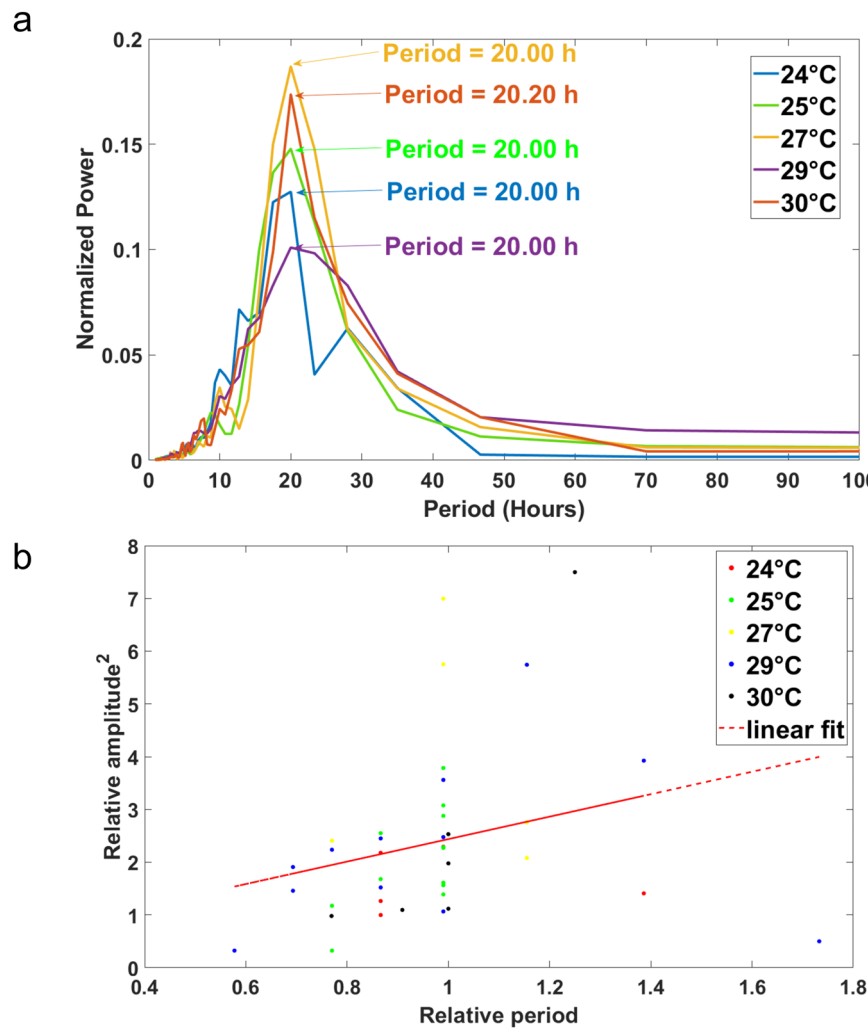

pattern"[7] obtained in real systems, the same peak at time t for different spatial x. The model also produced oscillations in genetic network components, such as FRQ (Fig. 11)[41]. Similar results were obtained with clock parameters determined from single cell data[27]—the significant difference is that models fitted to single cells did not produce oscillations in the signaling molecule in the medium (Fig. 11h).

One common feature of the clock components is the rising mountain range adjacent to the edge of the wavefront (Fig. 11e–g). This is due to the creation of new nuclei at the edge of the advancing wave (Fig. 11a). As these new nuclei get created, mRNAs and proteins also get synthesized at the edge of the wave in the model, leading to the stable banding patterns in time, much like a race tube[7] seen in Figs. 11e–g. This wave front provides a new mechanism for long-distance synchronization (Supplementary Table S5), in which advection and diffusion cause local mixing, but long-distance synchronization is achieved by carrying clock mRNAs and proteins in the advection wave.

Furthermore, we investigated whether synchronous behavior is captured in the phase trajectories of the fluorescent strain by analyzing the instantaneous phase using a Hilbert transform[32] (see Materials and methods). We used a band-pass filter for the experimental fluorescence trajectories to isolate the actual clock signal from low and high-frequency noise. We examined the three segments' fluorescence trajectories across the same set consisting of 6 serpentine channels. We took an average of each segment to create an equivalent of the hyphal clock model. The experimental results were then compared to the predictions by our hyphal clock model. Figure 12a, b illustrates the detrended experimental fluorescence as well as the model results. Clear oscillations are observed in both scenarios. We can also

observe that the segments were highly coherent (Fig. 12c, d), and the number of cycles corresponded well with the length of days. In Supplementary Fig. 9, the number of cycles for each segment is plotted out in real-time, consistent with the advection wave-producing phase synchronization. Interestingly we can observe a phase shift while observing different segments experimentally and in the model.

A final issue for the model is whether a combination of advection and diffusion called Taylor dispersion could lead to longer distance mixing at septal pores[42]. A precursor for Taylor dispersion is that there be differences in the velocity of nuclei at the edges and those in the middle of a filament, as might be seen in the currents in a tidal river emptying into the ocean. The effect of higher velocity flow in the center of the channel has the effect of pushing a zone of higher velocity and mixing as the water leaves the river and empties into the ocean. This is an example of Taylor dispersion. The drift velocities of nuclei were measured on the edges of the filament and in the center for four hyphal tips, and there is no significant difference in velocities of nuclei dependent on their location relative to the center of the filament (Supplementary Table S6). This observation is consistent with the findings of others examining Taylor dispersion in *N. crassa*[43].

## Discussion
The current study provides us with substantial insight into a relatively neglected area of cell synchronization biology. Previously, we were able to show synchronization at a macroscopic limit for single cells in a big chamber microfluidic device[27]. We also observed synchronization of cells at a much lower density[27] with a microwell device. Despite the increasing evidence for phase synchronization of cells, these were limited to *N. crassa* cells that were

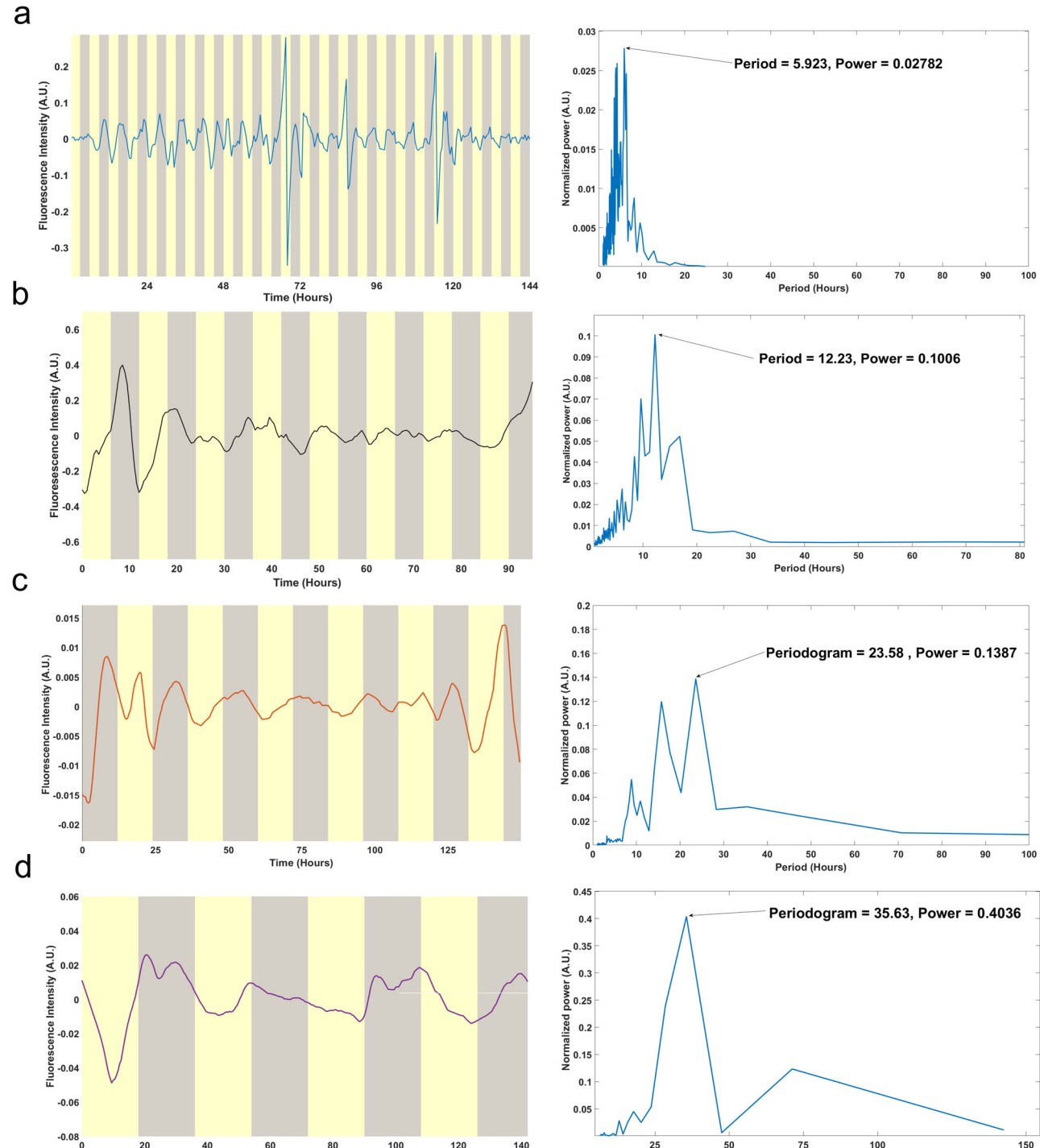

**Fig. 9 | Entrainment of MFNC9 filaments by light in *N. crassa* hyphae in serpentine channels to a 6, 12, 24, and 36-h day. a** Left: Average of 6 single hyphae trajectories detrended under a 3-h L/D regimen. Right: Average periodogram of hyphae of 6-h day length. **b** Left: Average of 9 single hyphae trajectories detrended under a 6-h L/D regimen. Right: Average periodogram of hyphae of 12-h day length. **c** Left: Average of 16 single hyphae trajectories detrended under a 12-h L/D regimen. Right: Average periodogram of hyphae of 24-h day length. **d** Left: Average of 3 single hyphae trajectories detrended under an 18-h L/D regimen. Right: Average periodogram of hyphae of 36-h day length. The FT method was utilized.

prohibited from growing. Here we established for the first time that hyphae have individual clocks with the capability of circadian rhythms, light entrainment, and temperature compensation as a prelude to providing evidence for filament synchronization.

We continued by asking whether the clocks in single hyphae synchronize under conditions of hyphal growth with sufficient media. We designed the serpentine microfluidic device (Fig. 1) to allow precise control

of the cellular environment with a fixed media flow rate. We demonstrated the existence of clocks for individual hyphae in a microfluidic chip (Supplementary Table S2 and Fig. 9). The width of the serpentine channel opening was determined to restrict the number of hyphae growing into channels. This allowed us to track single hyphae without much difficulty. We could observe clear oscillations that were comparable to luminescent bands tracked in race tubes[7] (Fig. 2), but these oscillations were noisier than in race

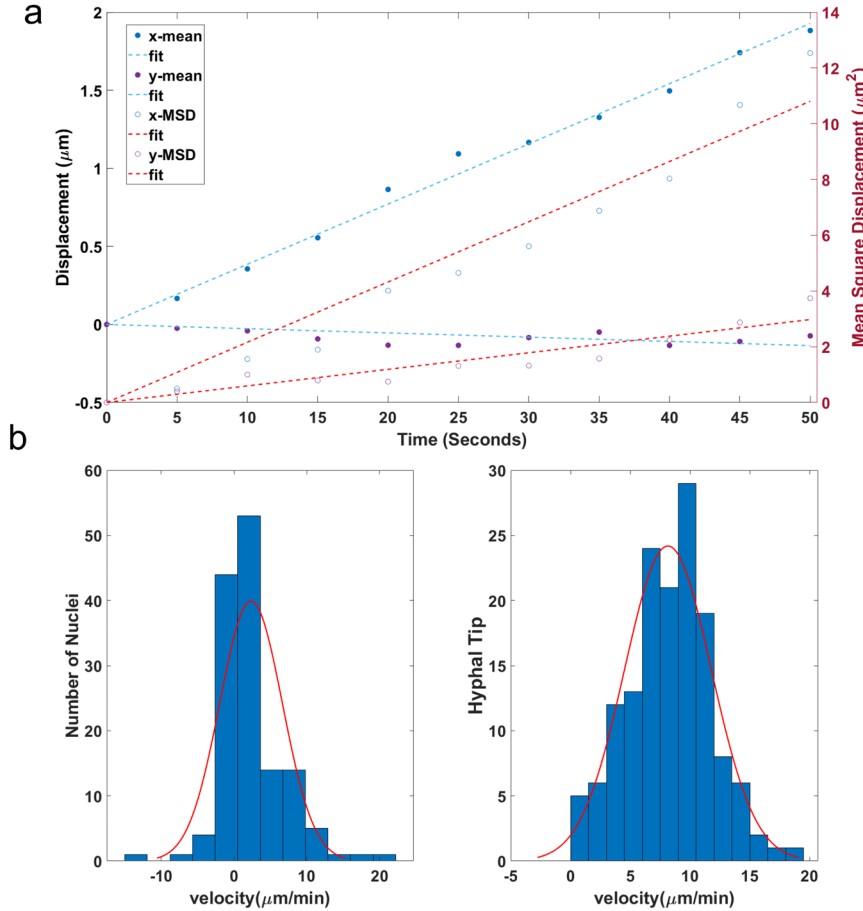

**Fig. 10 | Nuclear drift velocity and the Mean Square Displacement at the hyphal tip in the x-direction increase linearly with time using N2218-3 expressing a histone H1-GFP. a** Graph of nuclear drift velocity and diffusion velocity on the relative time. The $x$–$y$ coordinates of each nucleus (139 nuclei) were tracked over 11 sequential images (taken at 5 s intervals) and plotted against time. Each point in the graph is plotted with the mean displacement traveled from the origin over 11 sequential images. The drift velocity extracted from the slope of the straight-fit line calculated in the x-direction was 2.316 ± 16.6946 − 04 (2SE) µm/min. The results of the Pearson correlation were $r$ = 0.9961 (Fishers $z$ = 3.1146, $P$ = 9.75e − 11), and the Spearman rank correlation ($r_s$) was $r_s$ = 1 ($P$ = 0). The drift velocity in the y-direction was −0.162 ± 13.5826 − 04 (2SE) µm/min. Pearson correlation ($r$) for the latter was $r$ = −0.5509 (Fishers $z$ = −0.6197, $P$ = 0.0790), while the Spearman rank correlation ($r_s$) was $r_s$ = −0.5364 ($P$ = 0.0936). The diffusion coefficient in the x-direction was half of 12.97 ± 0.02 (2SE) µm²/min. The Pearson correlation was $r$ = 0.9833 (Fishers $z$ = 2.3872, $P$ = 5.8153e − 08), and the Spearman rank correlation ($r_s$) was $r_s$ = 1 ($P$ = 0). In the y-direction, the diffusion coefficient was half of the slope, 3.57 ± 0.0094 (2SE) µm²/min. The Pearson correlation was $r$ = 0.9376 (Fishers $z$ = 1.7181, $P$ = 2.02e − 05), while the Spearman rank correlation ($r_s$) was $r_s$ = 0.9636 ($P$ = 0). Standard errors were calculated by bootstrap resampling of mean displacement and mean square displacement traveled from the origin, respectively, for drift velocity and diffusion coefficient. The bootstrap sample size used was 1000. **b** Left: A histogram of the nuclear drift velocity profile for 187 nuclei was plotted. Right: Histogram of the growth velocity of the hyphal tip for 147 time points.

tubes (Supplementary Table S1). An interesting result seen while tracking luminescence in race tubes is that we observe synchronization over long distances of 30–60 mm (Fig. 2). Another question is whether or not synchronization varied with the number of filaments within a channel. One way to address this question is to design a new device that allows two adjacent channels of varying width to assay filament synchronization. Filaments in adjacent channels could be assayed for synchronization while varying the width of a channel could be used to control the number of filaments.

To characterize the physical processes underlying hyphal synchronization, a direct comparison can be made between the role of advection and diffusion in maintaining phase synchronization in fungal filaments. In 21 h (one period of the clock), the diffusion coefficient is half of 12.97 $\mu m^2/$ min or $\sqrt{16,342}$ = 128 $\mu m$, while the distance traveled under advection is 2.316 $\mu m/$ min or 2918 $\mu m$ in 21 h (Fig. 10). What is striking is when we turn off the diffusion process in the hyphal clock model, regular synchronization along the channel is still observed. This observation sparked the hypothesis that the model can generate synchronization solely with advection accompanied by growth through cell division. As nuclei and other molecules are swept along behind the growing fungal tip in the growth wave (within the advection zone), the clock mechanism components are also swept along as well to produce the synchronization observed spatially (Supplementary Table 5). As a result, processes such as advection and diffusion, operating locally to produce synchronization, get carried on a wavefront of growth (Fig. 11e–h), much like calcium spikes in other systems[19]. This theory of synchronization could explain how clock mutants perturb phase synchronization. The stable banding patterns generated by the wavefront (Fig. 11e–h) help to explain why banding patterns are seen on the macroscopic scale with race tube assays[7]. Attaining this result constrains the size of the advection zone at the beginning and ending time of the experiment. Without these cutoffs, it is difficult to achieve stable banding patterns. The cutoffs, however, do not constrain the ensemble of rate coefficients for the clock network—they are a feature of the physical process of advection. Besides that, the genetic network parameters are highly constrained by previous experiments[24,27]. Further research should be conducted to better identify which phase synchronization hypothesis is supported. This can be done by introducing cell perturbation systems[44–47], for example, time-dependent cyclic perturbations or varying media flow diffused into the

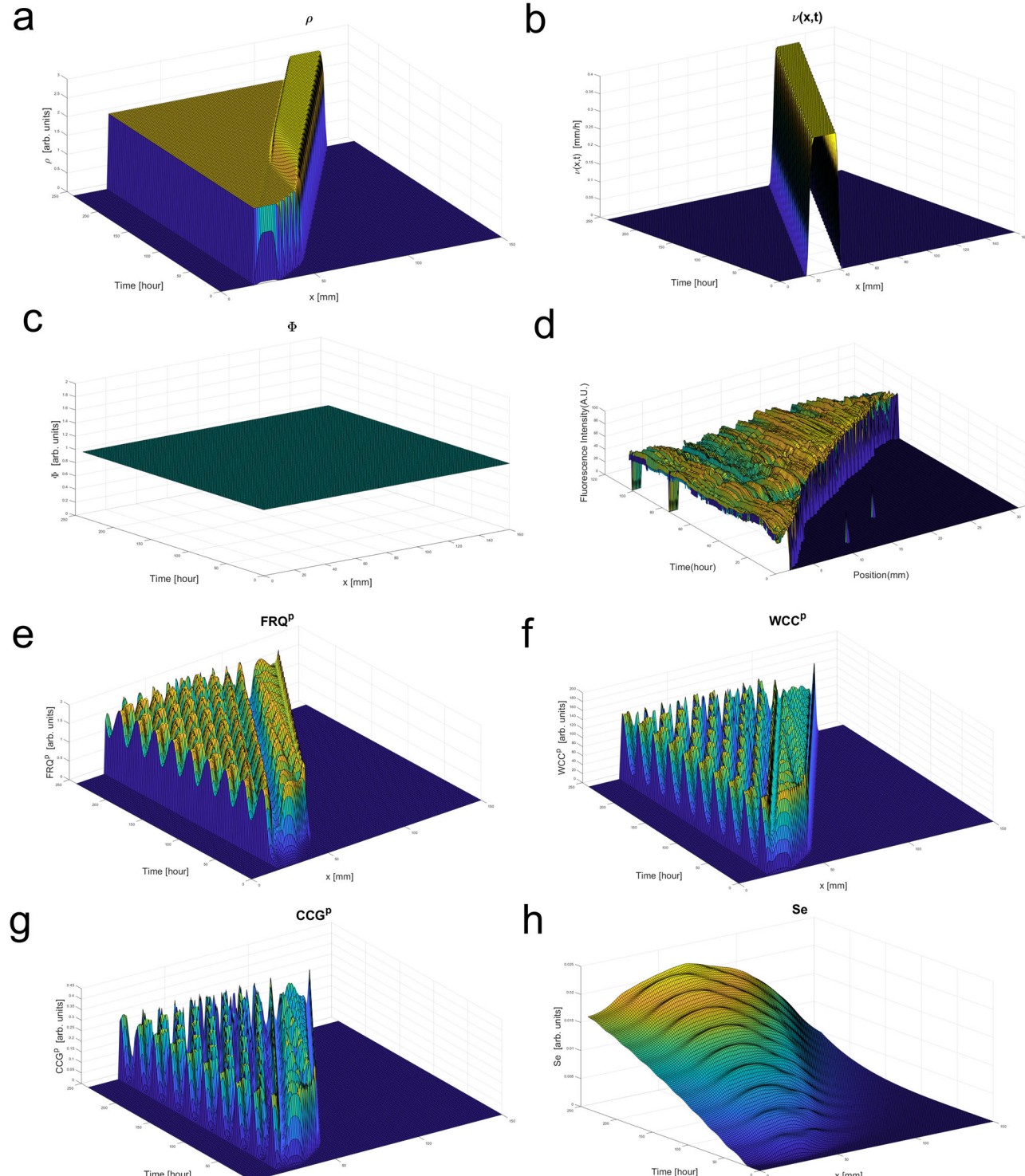

**Fig. 11 | The growth model specified the nuclear density, velocity profile, and food distribution as a function of spatial location ($x$) and time ($t$).** The hyphal clock model displays the "bands" observed in race tubes and the periodicity of the serpentine device in several key clock components including FRQ. **a** Nuclear density $\rho(x, t)$; **b** Initial velocity profile $v(x, t)$ for the growing hyphae(s); **c** food distribution $\Phi(x, t)$; **d** Observed experimental fluorescence profile over 32,000 µm or 32 mm in an individual serpentine channel. Figure 7a is only a 1.986 mm subset of (**d**). **e** The FRQ protein concentration is graphed as a function of spatial location ($x$) along the device and time ($t$); **f** the WCC protein concentration is graphed as a function of spatial location ($x$) along the device and time ($t$); **g** CCG protein usually observed in the device is also graphed as a function of spatial location ($x$) and time ($t$); **h** the communication signal in the media ($S_e$) is graphed as a function of spatial location ($x$) and time ($t$). The solutions of these partial differential equations are described in the Materials and methods.

serpentine chip[18] with slight modifications. Additionally, a new microfluidic device needs to be fabricated to allow hyphae that are entrained at different phases to synchronize with each other. We can observe the phase before and after two cell populations come in contact.

In the hyphal clock model presented, nuclei are deposited and proliferate along the hyphal growth front, also known as the advection zone, as seen in Fig. 3. This agrees with the concept that nuclear division requires nutrients and hence will deplete food behind the hyphal growth front. An

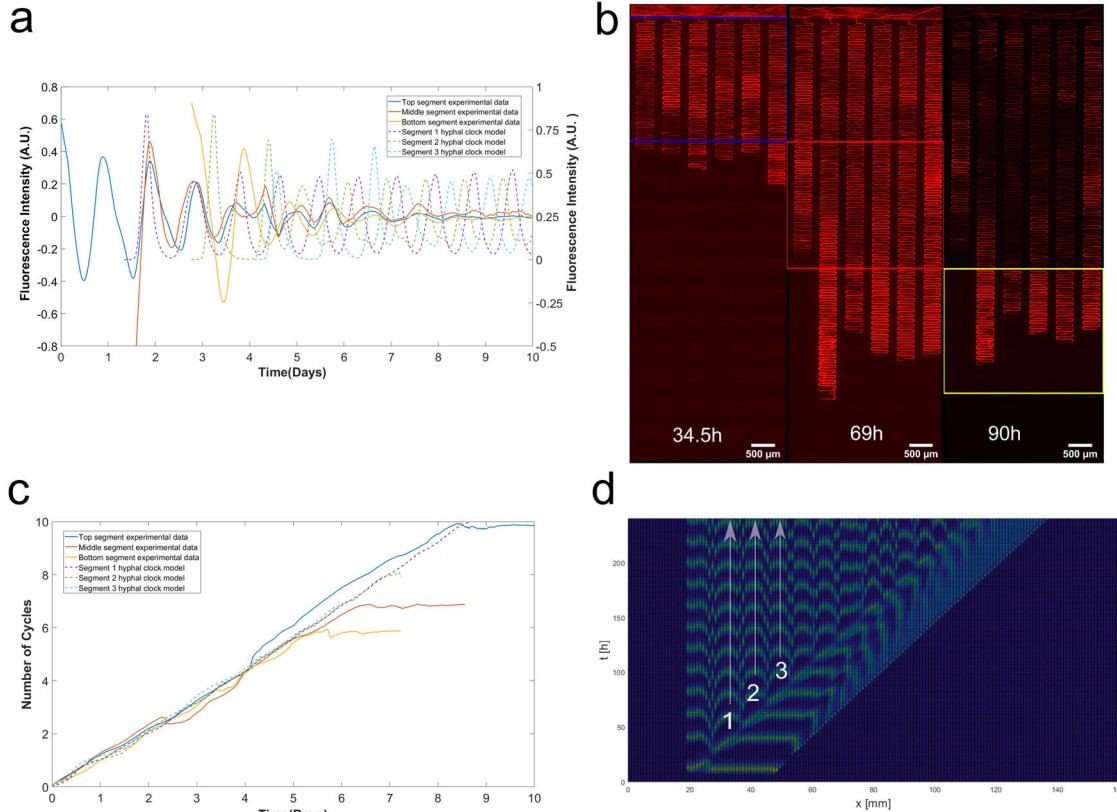

**Fig. 12 | Hilbert phase curves from experimental and model prediction.** Our hyphal clock model well predicts the Hilbert Phase curves of different segments behind the growing hyphal tip—the overlay of the Hilbert Phase curves indicates phase synchronization between fields of view. **a** Comparison of the experimental data and the model prediction for the obtained fluorescence signal for three segments. **b** Three representative fluorescence experimental images with the cropped segments, each containing 6 serpentine channels labeled with the colors corresponding to the curves shown in (**a**) and (**c**). **c** Plot of the experimental and model results of the number of cycles vs. time. **d** Three representative segments were obtained from the 2D image of the model corresponding to the curves shown in (**a**) and (**b**).

alternative hypothesis is that nuclei division occurs wherever food is present along the filament, accompanied by the process of mixing, which will result in the uniform distribution of nuclei along the filament[21,36]. While at first sight, bidirectional flow can be observed (Supplementary Movie S1), the estimated diffusion (Fig. 10) is not sufficient to establish this mixing as it would need to invoke an active transport mechanism. A second piece of evidence against this alternative hypothesis is that there is limited forward movement of nuclei observed away from the hyphal growth front outside the advection zone (Supplementary Movie S2). This supports prior reports that observed the aging of the septa[37,38]. A further test of the alternative hypothesis on nuclear division might begin by modifying the model to observe the effects of allowing nuclear division at locations with nonzero food density.

The synchronization hypothesis proposed here is that synchronization is local, and molecules involved in synchronization are swept along on the growth wave behind the growing hyphal tip, much like an action potential in a nerve. The Hilbert Transform is used to compute the analytical signal of the fluorescence intensities, which will then provide us with the instantaneous phase[32]. Suppose we examine the Hilbert phase at different segments along the filament behind the growth wave. In that case, the synchronization model predicts a strong correlation in the Hilbert Phase Curves for different segments of hyphae in the model and the serpentine microfluidics device. This is actually what is seen (Fig. 12). The measured Hilbert Phase curves for varied segments overlay nicely on the predicted Hilbert Phase curves of the model. Detrended trajectories (observed and expected) are shown in Fig. 12a, and the phase in cycles are plotted as a function of in Fig. 12c.

Our fluorescence trajectories demonstrated an increasing trend in fluorescence intensity at the U-turn sections of the serpentine channel (Fig. 7). We hypothesize that these could result from a higher possibility of branching around the U-turns due to the device geometry. Studies have used time-lapse live cell imaging of *N. crassa* in a maze-like microfluidic structure to observe how constraining geometries determine fungal growth[48]. They observed a hit & split phenomenon that would occur instantly after the contact between a hypha and a constraining structure. Recently, a report was made that *N. crassa* is a fast-growing fungus that possesses the ability to cover newly available nutrient-rich space[49]. However, when placed in spatially confined areas with obstacles, this may cause multiple new polarity axes to form, resulting in branching to occur[49]. An alternative hypothesis is the possibility of a change in the diffusion rate for hyphae growing in the U-turns. A decrease in the diffusion rate may allow branching to occur more frequently around the channel's curved area than straight channels. To test these hypotheses, we can fabricate a microfluidic chip with varying geometries parallel to each other on the same device for simultaneous tracking.

The model does incorporate some form of local communication internal or external to a filament. Some information on the quorum sensing signal is now available between spores. Previous microfluidics experiments suggest the quorum sensing signal has a spherical radius of no more than 13 nm and is density dependent[27]. Furthermore, previous work in fungi would suggest consideration of aromatic alcohols as candidate quorum sensing signals[3]. One of the predictions of the model is that the quorum sensing signal oscillates in the dark (Fig. 11g).

The model presented has both some successes and limitations. The striking success is producing stable banding patterns with peaks at time *t* for all spatial *x* with the right period. This provides an explanation for the

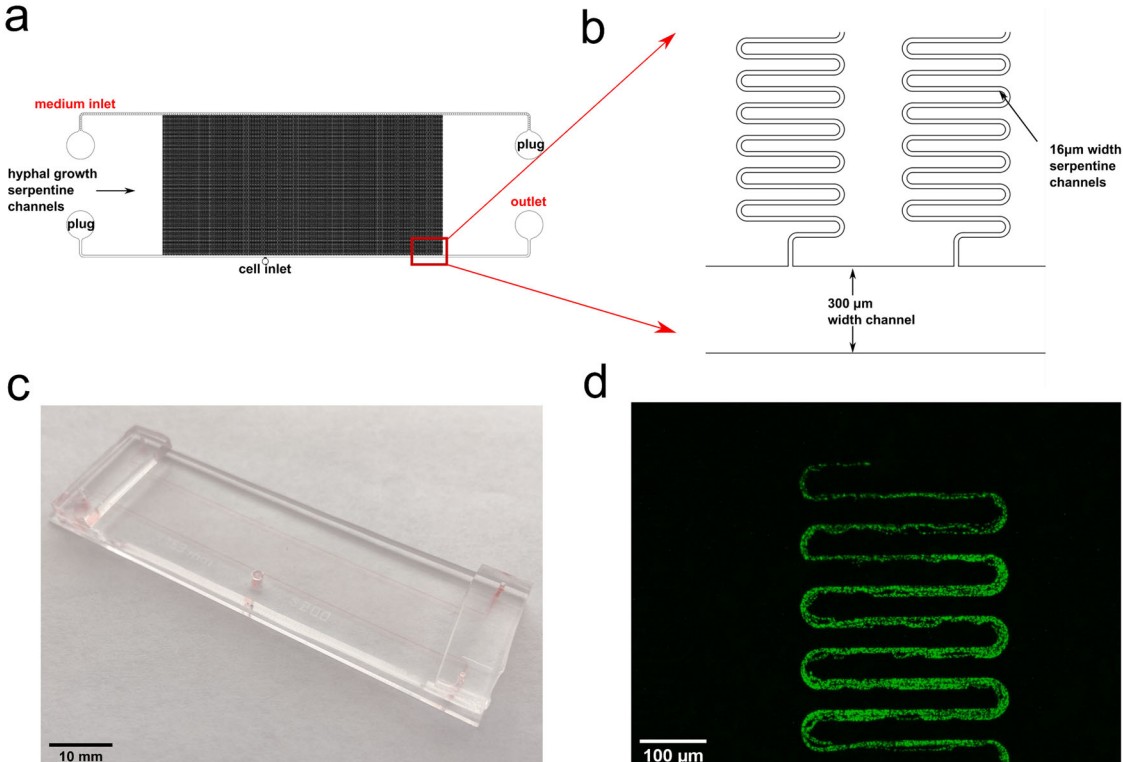

**Fig. 13 | Microfluidic platform designed to track hyphae growth.** Serpentine microfluidics device allows observation on 1–95 hyphae over 6 days. **a** Full schematic of the serpentine chip presenting the whole set-up of the experiment. **b** Magnified schematic showing the growth channels with 300 μm width and serpentine channels for hyphae growth is 16 μm wide and 198,241 μm long. The dimensions of the serpentine channel were chosen to accommodate only a few hyphae. A hypha can range from 8–15 μm[30]. **c** An image of the serpentine chip. The device's dimensions are 65,500 μm × 20,150 μm and features 95 parallelized serpentine channels. A cell inlet port with a 2 mm diameter is located in the middle of the larger medium inlet channel. **d** Visualization of a *N. crassa* strain N2218-3 expressing a histone H1-GFP growing along the serpentine channels.

banding patterns in race tubes as well[7]. The phase curves for different segments are also well predicted by the model (Fig. 11c). On the other hand, there are consistent phase differences between apical, subapical, and basal parts of the hyphae needing explanation. One way to accommodate these differences is to allow for septal occlusion as hyphae age[39]. These will allow additional dispersion of flow at septae. The model created is quite general and designed to accommodate different microfluidic geometries to be used in the future. It is also set up for iterative improvement by future experiments with ensemble methods of systems biology[24].

Another limitation of the model is the effects of intrafilament vs. interfilament communication within the model. It would still be desirable to distinguish both forms of communication experimentally, and one approach to this problem is the design of new microfluidic devices to test the modes of communication. Devices are being constructed to filter for small signaling molecules[50].

This study illustrates a high-throughput serpentine chip to demonstrate the existence of cellular clocks in individual hyphae, the dominant life stage of filamentous fungi. Hyphae within the device are also phase-synchronized with respect to their hyphal clocks. The device used provided multiple serpentine channels to facilitate controlled hyphal growth of *N. crassa*. Trapped hyphae successfully grew and elongated across the serpentine channels. The resultant hyphae grew down narrow growth channels that enabled measurements on their fluorescence intensities to study the clock. To our knowledge, this is the first report of the existence of cellular clocks in growing individual hyphae with a supporting hyphal clock model with similar behavior. The hyphal clock model has many of the qualitative properties of filaments in channels and suggests that the wave of growth itself traveling down the channel could lead to phase synchronization of the syncytium. The platform has the potential to aid us in our understanding of the growth process of *N. crassa* and other fungal species[48,51]. It provides a broadly applicable research tool to introduce perturbations for cells to study fundamental processes, such as the clock, linked to growth.

## Materials and methods
### Fluorescent strains
For observation of the clock strain *bd,ccg-2P:mCherry* was used[26] (MFNC9), where *ccg-2P*: *mCherry* denotes a *clock-controlled gene-2 Promoter* region (*ccg-2P*) fused in frame with a mCherry reporter. The fluorescent strain mat A his-3 + ::Pccg-1-hH1 + -sgfp+[30] (N2218-3) was used to observe nuclei.

### Microfluidics device design and fabrication
The microfluidic device (Fig. 13) contains a cell loading channel with an inlet to load an agar plug, one medium loading channel, and 95 serpentine regions. Serpentine channels resemble those in other devices, allowing hyphae to grow several days[17,52] within the device. The device features channels of two varying heights, where the medium loading channel (30 μm), and serpentine regions (15 μm) were fabricated with two photomasks. It consists of four inlets, one inlet to deliver media and a medium outlet. At the same time, the other two have plugs attached to ensure a constant flow of media across the serpentine channels. 15 μm serpentine regions were added with SU-8 3010 (MicroChem, Westborough, MA). The second step was to have 30 μm medium loading channel fabricated with SU-8 2025 (MicroChem, Westborough, MA). Channel heights were measured by a profilometer (Veeco Instruments, Chadds Ford, PA). Prior to PDMS casting, the fabricated master mold was vapor covered with (tridecafluoro-1,1,2,2-teterahydroctyl) trichlorosilane for 15 minutes and subsequently heated to 120°C for 10 min. The microfluidic device was built with PDMS (10:1 w/w) by a standard PDMS replica molding technique[53]. Polydimethylsiloxane (Dow Corning, Midland, MI) was poured onto the silicon wafer, baked at 80 °C for 2 h, and bonded on a glass slide to produce the final device.

## Inoculation of agar plug into serpentine microfluidics device

In a typical procedure, all microfluidic channels were initially primed with 0.1% glucose media to ensure the reduction of bubbles. The prepared agar plug with 1-day-old conidia is placed in the central cell inlet and sealed with a 3D-printed plug. A 3 mL syringe with 0.1% glucose media was constantly perfused into the medium infusion channel and flowed from the outlet (Fig. 13) throughout the time-lapse experiment. We connected the waste outlet to a conical tube for collecting excess medium. The device was then placed under constant light exposure (5370 lux) for at least 2 h before placing it in the dark for imaging. Hyphal growth generally occurs after a period of 6 h until they begin to grow into the nearest serpentine channels. After required light exposure, the serpentine device was placed on the microscope stage.

## Image acquisition

Live cell imaging was done with a CCD camera (AxioCam HRm, Carl Zeiss Microscopy, LLC, Thornwood, NY) to record the fluorescence intensity of the cells through a microscope (Axio Imager M2, Carl Zeiss Microscopy, LLC, Thornwood, NY) with a motorized x-y stage (Mechanical stage $75 \times 50$ R, Carl Zeiss Microscopy, LLC, Thornwood, NY) in a dark room. The excitation light used was guided through a filter set (Filter Set #61HE, Carl Zeiss Microscopy, LLC, Thornwood, NY) at 585/35 with an emission at 645/60. Images were taken every 30 min with an exposure time of 1200 ms. Images were stitched and exported as an 8-bit grayscale to accommodate the large file size because of the length of the time-lapse experiment. Zeiss A2 inverted microscope was used to observe calcofluor labeling with the corresponding filters.

## Processing of hyphal images

Fluorescence images were tracked using a pipeline built with CellProfiler. Images were first converted to 8-bit files and cropped based on the segments to analyze to accommodate for the processing time and the size the software can handle. Custom Python codes were written for each experiment to extract the required data. Data were then exported to MATLAB. Custom MATLAB scripts were utilized to produce periodograms or fluorescence trajectory plots. Each fluorescence trajectory was detrended[2] with a 24-h moving average. Their periodograms were computed.

## Velocity profile, drift, and diffusion velocities of the hyphal clock model

*N. crassa* strain N2218-3 expressing a histone H1-GFP provided us with a method to track the movement of nuclei. CellProfiler[54] allowed tracking of nuclei by obtaining their x-y coordinates. Nuclear drift velocity, as well as diffusion velocity were obtained by extracting the x-y coordinates of each nucleus (139 nuclei) that were tracked over 11 sequential images (taken at 5 s intervals) and plotted against time. Each point in the graph is plotted with the mean displacement traveled from the origin over 11 sequential images. The drift velocity was extracted from the slope of the straight-fit line. The diffusion coefficient is obtained from the slope of the mean square of displacement from the nucleus tracked. The growth velocity of the hyphal tip is obtained by the displacement of the x-y coordinates extracted with ImageJ from bright-field images taken at 10 s intervals.

## Processing of race tube time-lapse movie

The race tube data on *frq-luc-I* were downloaded for Supplemental Movie File 4 at https://ec.asm.org/content/7/1/28/figures-only[7]. A python program(vti) was written to convert the video to an image sequence, which was then processed as with the hyphae images.

## Measuring signal-to-noise ratio

The experimental data's estimated mean amplitudes of signal and noise from the experimental data were applied to calculate the signal-to-noise ratio for the fluorescent intensity signals of hypha and race tube data.

The built-in smooth function in MATLAB was applied to the original experimental data to estimate the signal, and the window size of the smooth function was 5 hours. The mean amplitude was calculated using the first three days of signal data. Then the noise was estimated by taking the absolute values of the difference between the original data and the estimated signal. Then the mean amplitude of the noise was calculated using the first three days' noise data. The signal-to-noise ratios of the hyphae's fluorescent intensity signals were then calculated using the mean amplitudes of the signal divided by the mean amplitude of the noise.

## Generation of white noise for Kuramoto *K* calculation

The noise model generates 127 trajectories, and each trajectory would have 480 data points. All initial values of the trajectories are set to 0, and for each corresponding step, white noise is added to the trajectory respectively. The trajectories were detrended with a 24-h moving average over time (Supplementary Fig. S10). The synchronization was calculated among the trajectories and Kuramoto *K* is 0.53.

## Calculating phase curves

We first apply a band-pass filter to our fluorescence time series data. The band pass filter consists of two steps: (1) Detrending[2] with a moving average with a longer time window (greater or equal to the daily period) (2) An additional smoothing step with a shorter time window (smaller than the daily period). Next, to calculate phase for a detrended fluorescent series $x(t)$, the Hilbert transform $\tilde{x}(t) = PV \frac{1}{\pi} \int_{-\infty}^{\infty} \frac{x(\tau)}{t-\tau} d\tau$ was computed from the Fast Fourier Transform[55] of $x(t)$. The Hilbert phase $F^H(t)$ is defined as the phase angle between the Hilbert Transform $\tilde{x}(t)$ and $x(t)$ by $F^H(t) = \tan^{-1}(\frac{\tilde{x}(t)}{x(t)})$. The Hilbert Phase was continuated to $F^C(t)$ to avoid discontinuities in the phase angle at $\pi$ and $-\pi$. The continuization was done recursively through the relation: $F^C(t+1) = F^C(t) + m^C(t)2\pi$, where at each step the argument m was chosen to minimize: $Df_m = \left| F^H(t+1) - F^C(t) + 2\pi m \right|$. With the continuized Hilbert Phase $F^C(t)$, the phase is defined by: $M^C = \left\lfloor F^C(t_1) - F^C(t_0) \right\rfloor / 2\pi$ in units of cycles. An accessible description of these phase measures and code to calculate them in MATLAB are available[32] with associated MATLAB in GitHub.

## Hyphal clock model

A simulator of the hyphal clock model is written in MATLAB. The script with output is given in Supplementary Data 1. We provide a manual for using the simulator in Supplementary Data 2. A full description of the model is given in Supplementary Data 3. The parameters for the clock model used are provided in Supplementary Table S7. We also included another set of parameters that is provided in Supplementary Table S8.

## Solving the hyphal clock model

The partial differential equation (PDE) system is converted into an ordinary differential equation (ODE) system by a discretization of the x-coordinate on an equidistant x-grid, $x_n$. Specifically, if a given species of concentration $f(x, t)$ is subject to transport by diffusion inside the filament, the 2nd x-derivative of $f(x, t)$, enters into the PDE, *via* the corresponding diffusion term, and it is approximated by

$$\partial_x^2 f(x_n, t) \approx \frac{1}{(\Delta x)^2} \left[ f(x_{n+1}, t) + f(x_{n-1}, t) - 2f(x_n, t) \right]$$

Here, $\Delta x = L/N_L$ is the x-grid spacing, $L$ is the length of the x-domain, $[0, L]$, and the x-grid points are $x_n = n\Delta x$ for $n = 0, 1, \ldots, N_L$ for a chosen grid size $N_L$.

Similarly, if a given species of concentration $f(x, t)$ is subject to advection inside the filament, with a drift velocity field $v_f(x, t)$, the species is transported by a concentration drift or advection current, given by

$$J_f(x, t) = v_f(x, t) f(x, t)$$

The 1st $x$-derivative of $J_f(x, t)$ then enters into the PDE, via the corresponding advection term, and it is approximated by

$$\partial_x J_f(x_n, t) \approx \frac{1}{\Delta x}\left[\left|J_f(x_n, t)\right| - R\left(J_f(x_{n-1}, t)\right) - R\left(J_f(x_{n+1}, t)\right)\right]$$

where $R(J) := \max(J, 0)$ for real argument $J$.

The foregoing approximations to the $x$-derivatives preserve the particle number conservation laws that are obeyed by the respective diffusive and advective transport processes described in the underlying exact PDE system.

The resulting ODE system for the corresponding discrete dynamical variables, $f_n(t) \equiv f(x_n, t)$, is then solved by an adaptive 4th order Runge–Kutta algorithm, over a solution time interval, $[0, T]$, of duration $T$, with an ODE solution relative error tolerance of $10^{-6}$ or better. To estimate and control the numerical errors introduced by the discretization of the $x$-coordinate, the ODE system is solved for a doubling sequence of $x$-grid sizes $N_L$, i.e., first for some initial grid size $N_L^{(0)}$, then for $N_L^{(1)} = 2N_L^{(0)}$, then for $N_L^{(2)} = 4N_L^{(0)}$ and so on. This is done until the relative change in the PDE solution values $f(x, t)$ during the (last) $N_L$-doubling step changes by less than some maximum allowed relative error for each species concentration $f$. That is, we quantify the numerical discretization error in terms of the relative root mean square (RMS) change of $f(x, t)$ from grid size $N_L^{(\nu)}$ to $N_L^{(\nu+1)} \equiv 2N_L^{(\nu)}$, defined by

$$r_f^{(\nu)} := \frac{\mathrm{RMS}_\nu\left[f^{(\nu+1)} - f^{(\nu)}\right]}{\mathrm{RMS}_\nu\left[f^{(\nu)}\right]}.$$

Here, $f^{(\nu)}(x, t)$ denotes the ODE solution obtained for some species concentration $f(x, t)$ on an $x$-grid of size $N_L^{(\nu)} = 2^\nu N_L^{(0)}$. For any function $g(x, t)$, the quantity $\mathrm{RMS}_\nu[g]$ denotes the root mean square of $g(x, t)$, over the solution time interval $[0, T]$ and over the $x$-grid of size $N_L^{(\nu)} = 2^\nu N_L^{(0)}$, defined by

$$\mathrm{RMS}_\nu[g] := \sqrt{\frac{1}{N_L^{(\nu)}} \sum_{n=0}^{n=N_L^{(\nu)}} \frac{1}{T} \int_0^T \left|g\left(x_n^{(\nu)}, t\right)\right|^2 dt}.$$

The $x_n^{(\nu)} = \left(L/N_L^{(\nu)}\right)n$ for $n = 0, 1, \ldots N_L^{(\nu)}$ are the $x$-grid points for grid size $N_L^{(\nu)}$. Note that the ODE solution values $f^{(\nu+1)}$, obtained for grid size $N_L^{(\nu+1)} = 2N_L^{(\nu)}$, are also defined, i.e., numerically calculated, for all $x$-grid points $x_n^{(\nu)}$ of the smaller grid of size $N_L^{(\nu)}$, since the grid points $x_n^{(\nu)}$ are a subset of the grid points $x_n^{(\nu+1)}$. The difference function, $f^{(\nu+1)} - f^{(\nu)}$, entering into the definition of $r_f^{(\nu)}$, is therefore also defined for all $x$-grid points $x_n^{(\nu)}$.

The $t$-integrals in the foregoing definition of $\mathrm{RMS}_\nu[f]$ are approximated by applying the trapezoidal numerical integration rule, with summation over an equidistant ODE solution output $t$-grid of grid spacing $\Delta t$. For the numerical results presented here, we used $\Delta t = 1h$ for solution time durations of up to $T = 240\,h$.

If we start from a sufficiently large initial $x$-grid size, $N_L^{(0)}$, the doubling sequence can be terminated already after one doubling step, i.e., at $\nu = 0$.

### Statistics and reproducibility
Statistical analyses were done in MATLAB and reported in the figure legends when necessary. All samples for analysis were collected with a minimum of three independent replicates.

### Reporting summary
Further information on research design is available in the Nature Portfolio Reporting Summary linked to this article.

### Data availability
The numerical source data and codes used to generate the figures for this paper are publicly available on GitHub at https://github.com/jc68175/serpentine. The software for the hyphal clock model is available as a supplement. All other raw data are available upon reasonable request to arnold@uga.edu.

### Code availability
The MATLAB codes utilized to generate the figures for this paper are publicly available on GitHub.

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

## Acknowledgements

This study was supported by the U.S. National Science Foundation (NSF), Division of Molecular and Cellular Biosciences under award numbers 1713746 and 2041546.

## Author contributions

J.H.C., X.Q., H.S., L.M., and J.A. conceptualized and planned the study. J.H.C. carried out data acquisition. Y.L., J.H.C., and L.M. carried out microfluidics device design. Y.G., J.H.C., and E.K. assisted with making agar plugs. S.B., H.S., and X.Q. carried out modeling. J.H.C. and X.Q carried out data and image analysis. J.H.C. and J.A. drafted the first version of the paper. All authors contributed to paper editing and reviewing.

## Competing interests

The authors declare no competing interests.
