## [Peer Review File · Communications Biology]

Reviewers' comments:

Reviewer #1 (Remarks to the Author):

Review of the Manuscript "The clock in growing hyphae and their synchronization in *Neurospora crassa*" by Cheong et al.

1. Overall impression of the work

The major, and really major, novelty of the paper is the study of autonomous clock in hyphae. To the best of my knowledge this has not been done, certainly not at this level of precision provided by the confining environment of microfluidics networks. This is excellent good work, with essentially very little to change – in my view.

2. Specific comments

3.1. General observations

3.1.1. The only major observation that I have is that, while I am convinced that the authors accurately account for the variation of temperature, I am not convinced that the same applies to the spatial distribution of nutrients, and chemical species in general. I understand that the authors provided a flow on media long the growing hyphae in the serpentine, but still some back of the envelope would help, as hyphae would have various nutrient uptake along their length. Now, that the authors used "flow around", would the surrounding pressure change the results? Perhaps in some other work.

3.2. Specific observations

3.2.1. Page 5. "Reducing the width of the channels may assist in the hyphal tracking process; however, we could optimize the hyphal tracking procedure, and branching did not affect our results." I quite do not get it. What was done, how do we know that it does not affect results, etc. Please rephrase for clarity, and perhaps, if needed provide additional information.

3.2.2. Figure 1 and surrounding narrative. I think that a more detailed description, perhaps with relevant micrographs, would do more justice to the work.

3.2.3. Minor comment: why is one movie a .mov, and the other an .avi?
To conclude I grade this submission as "Minor Revision".

Reviewer #2 (Remarks to the Author):

This study established a method/equipment applicable in investigating cellular clock at individual hypha level, using *ccg2* promoter driving fluorescent strain of *N. crassa* as an example. By this equipment, the authors were able to measure hyphae growth, nuclei movement, circadian rhythm within single hypha, over a period of 6 days, based on which they verified the hyphal clock model integrating properties of nutrients and quorum sensing, entrained by light and capable of temperature compensation. Overall, this work is novel and provided a useful tool for further studying circadian rhythm at single hypha level.

I have a few queries listed in the following:

1. Is it known what QS signal that affects *N. crassa* circadian rhythm? Does it also contribute in cellular synchronization of hyphae? Hope to see some discussion on this aspect.

2. Following the query 1, does such QS signal synthesis also oscillate in day/light cycle, besides cell-density dependent?

3. I am not sure whether I understand it right: Figure 5 shows that in a serpentine channel there are various numbers of hyphae (1-6). I wonder is the clock synchronization in the single hypha alone in a channel same or different from that of a single hypha present in a channel containing more than 1 hyphae?

Reviewer #3 (Remarks to the Author):

The team use serpentine microfluidic devices to monitor long time growth of *Neurospora crassa* hyphae, along with their maintenance of circadian rhythm (monitored via a *ccg-2*-reporter). Although microfluidic channels have previously been used (by the same group) to study circadian rhythm in conidia, and by Hong et al. for long time imaging of individual hyphae, their full potential has remained far from realized. So the experimental work done in this paper seems appropriate. Questions about the spatial extent of synchronized behavior are important to understanding the cell biology of this important model circadian system. Both experiments and modeling have, to date, elucidated temporal dynamics, while treating the organism as a single well-mixed compartment, and the data presented here go some way toward clarifying how nuclei and cytoplasmic cues enable different parts of the same organism to remain synchronized.

There is quite a lot of new experimental data, and I appreciated Fig 12A, especially, showing that three parts of a single hypha could maintain the same phase, despite being mm's apart. I also found the study of mean nuclear velocities along the hypha to be very interesting. I was surprised that the mean nuclear velocity in the subapical hypha is less than the speed of the growing tip, though I have some comments (below) on the importance of studying population variabilities there.

There are two main sources of revision that must be made to the paper before I can recommend it for publication:

1. At the outset, the paper makes clear that synchronization within hyphae can rely on the same mechanisms as conidia, or upon intracellular sharing of signals. However, the data don't really explain the extent to which this experiment can differentiate between the two types of signalling. It seems like the model is supposed to do this work (though I don't think it does). I'm curious whether there are any evidences of phase differences upstream versus downstream that could demonstrate that cues are following the cytoplasmic flow. Conversely if cues are diffusing outside of the hypha, then can the investigators use their perfusion mechanism to create directional flows of cues that change the pattern of synchronization?

2. The mathematical modeling does not seem to me to be successful or well-considered. Reaction-advection equations are presented in the main text (I remark here that there should be a minus sign in front of the divergence of F term in equations 1 and 6), promising a relatively straightforward modification of existing well-mixed (ODE) models to include nuclear populations, nutrients and their transport. However, when we read the supplement, we discover that many extra ingredients are added for intracellular and extra-cellular transport, along with an additional signaling mechanism (with intracellular and extracellular concentrations $[Si]$ and $[Se]$), couplings to nuclear division rates (which also need to be parameterized to ensure a carrying capacity, etc).

Now, even the authors of the original well-mixed models would (hopefully) admit that their models, which contain many different species, reaction-rates etc., are not well-constrained by existing experimental data, and correspondingly contain many different rates that are enormously separated in their time-scales. Here, yet more complexity is layered on, giving the investigators immense extra freedom to choose parameters and initial conditions. Yet, the fit to real data is not good -- importantly, in the model, there are consistent phase differences between apical, sub-apical and basal parts of the hypha, that I do not see in the real experimental data (Fig 11a). Meanwhile, the 'good' agreement in Fig 11c, merely indicates that the model is correct on the period length of the circadian oscillator. Now, I want to emphasize that I do not expect models to perfectly agree with data, we learn a lot about the sufficiency of assumptions etc., when a simple model disagrees with experimental data. But this model is so complex, that its failure leaves us uncertain about which of the many assumptions are incorrect. For what it's worth, I suspect that the main problem is the lack of proper modeling of diffusion within the hypha. Diffusive terms are included in a few of the PDE equations. But transport typically produces significant Taylor dispersion. And in *Neurospora*, where protoplasmic flows must periodically focus down into narrow pores, small amounts of diffusion can lead to large dispersion of flows. Taylor dispersion is an odd phenomenon in that lowest diffusive elements (e.g. nuclei) can have highest

dispersivities.

As I write this, I think that the best approach the authors could take is to work with a very simple, qualitative model (e.g. see the work of Peskin on single feedback loop circadian rhythm models), and to specifically test the effect of varying diffusivity within (and outside of) the cell.

Additional structural issues:

The investigators should conform to best practices for data sharing, including all of their experimental data, along with the original matlab code used to solve the model (even experienced Matlab programmers will have problems understanding the printout of the code that is offered with the manuscript).

Non-major considerations:

The general types of algorithms used to solve the PDEs should be described in the text; this includes the form of discretization: that they are using method of lines, what their time integration looks like, how they are treating the finite difference approximations for the derivatives to avoid numerical instability, etc.

The authors note that there are three main mechanisms by which conidia can maintain synchronous oscillations: contact and diffusible QS and stochastic synchronization. Two papers reference stochastic synchronization, while Stan Liebler's paper seems to be about robustness against stochastic effects and seems less well located. The text says three additional mechanisms exist for hyphae, but I think only intracellular synchronization is actually distinctive to syncytia, cf. single fungal cells.

Replicate numbers are not made clear or uncertainties considered: is the important figure 11a based on a single hypha? How do other hyphae compare? How do the period uncertainties that come from the very noisy data (see Fig 6) affect the results?

Reviewers' comments:

Reviewer #1 (Remarks to the Author):

Review of the Manuscript "The clock in growing hyphae and their synchronization in *Neurospora crassa*" by Cheong et al.

1. Overall impression of the work

The major, and really major, novelty of the paper is the study of autonomous clock in hyphae. To the best of my knowledge this has not been done, certainly not at this level of precision provided by the confining environment of microfluidics networks. This is excellent good work, with essentially very little to change - in my view.

Response: We are grateful for your positive comments on the manuscript.

2. Specific comments

3.1. General observations

3.1.1. The only major observation that I have is that, while I am convinced that the authors accurately account for the variation of temperature, I am not convinced that the same applies to the spatial distribution of nutrients, and chemical species in general. I understand that the authors provided a flow on media long the growing hyphae in the serpentine, but still some back of the envelope would help, as hyphae would have various nutrient uptake along their length. Now, that the authors used "flow around", would the surrounding pressure change the results? Perhaps in some other work.

Response: This is a very nice suggestion to do some back of the envelope calculations of uniformity of flow in the device. This is done in **Fig. S1** using existing COMSOL simulation software. **Fig. S1b** shows the uniformity of the flow through the device. We do agree that as hyphae elongates there will be pressure buildup in the serpentine channels, causing media to be unable to flow throughout the device. For future work, we will fabricate a device to provide nutrients throughout the whole serpentine channel.

3.2. Specific observations

3.2.1. Page 5. "Reducing the width of the channels may assist in the hyphal tracking process; however, we could optimize the hyphal tracking procedure, and branching did not affect our results." I quite do not get it. What was done, how do we know that it does not affect results, etc. Please rephrase for clarity, and perhaps, if needed provide additional information.

Response: To make this sentence clearer, we have rewritten the sentence into several sentences.

Action: We have added the following clarifying text.

“The width of the chip was maintained at 16 μ m to ensure the possibility of media transported along the channels via diffusion to support a growing filament. Reducing channel width further, however, would adversely affect the flow of media through the channel, affecting filament growth differentially along the channel.”

3.2.2. Figure 1 and surrounding narrative. I think that a more detailed description, perhaps with relevant micrographs, would do more justice to the work.

Response: We have added more micrographs to **Fig. 1** and added more details in the description as suggested. Thank you so much for your comments. We added the following clarifying text for Figure 1.

“In step 1 of the protocol, media was loaded into the chip with a syringe pump continuously till the whole chip was filled. A *N. crassa* agar block with fluorescent strain MFNC9²⁶ was then loaded into the cell inlet port located at the lower central position as shown in **Fig. 13a**. To prevent the agar block from drying out, a 3D printed plug was used to seal the cell inlet port. In Step 2, the chip containing the agar block was placed under a light source with at least 2 hours of light at 5370 lux. After exposure to light, the microfluidic platform was placed onto the microscope stage located inside a temperature-controlled enclosure. Throughout each experiment, media was constantly supplied to the microfluidic platform with a syringe pump. In step 3, filament growth of *N. crassa* cells was viewed (micrograph on the left) by time-lapse fluorescence imaging with a time interval of 30 minutes. For specific temperature compensation experiments, we varied the temperature with the temperature control instrument while ensuring the device was maintained in the dark. Meanwhile, a LED light source was placed into the enclosure and used for light entrainment experiments while maintaining a constant temperature. Time series of fluorescence measurements over at least four days were obtained through the images collected.”

3.2.3. Minor comment: why is one movie a .mov, and the other an .avi?

Action: Thank you for pointing it out for us. We have made sure both movies are of the same format (.mov).

To conclude I grade this submission as “Minor Revision”.

Response: Thank you for your support and comments on the manuscript.

Reviewer #2 (Remarks to the Author):

This study established a method/equipment applicable in investigating cellular clock at individual hypha level, using ccg2 promoter driving fluorescent strain of N. crassa as an example. By this equipment, the authors were able to measure hyphae growth, nuclei movement, circadian rhythm within single hypha, over a period of 6 days, based on which they verified the hyphal clock model integrating properties of nutrients and quorum sensing, entrained by light and capable of temperature compensation. Overall, this work is novel and provided a useful tool for further studying circadian rhythm at single hypha level.

Response: Thank you for your interest and comments on the work.

I have a few queries listed in the following:

1. Is it known what QS signal that affects N. crassa circadian rhythm? Does it also contribute in cellular synchronization of hyphae? Hope to see some discussion on this aspect.

Response: Adding a discussion of the QS signal is a great suggestion. We have followed your suggestion. Thank you very much.

Action: We have added the following paragraph on the QS signal to the discussion:

“The model does incorporate some form of local communication internal or external to a filament. Some information on the quorum sensing signal is now available between spores. Previous microfluidics experiments suggest the quorum sensing signal has a spherical radius of no more than 13nm and is density dependent¹. Furthermore, previous work in fungi would suggest consideration of aromatic alcohols as candidate quorum sensing signals². One of the predictions of the model is that the quorum sensing signal oscillates in the dark (**Fig. 11g**).”

2. Following the query 1, does such QS signal synthesis also oscillate in day/light cycle, besides cell-density dependent?

Response: In **Fig. 11g** the QS signal does oscillate in D/D conditions. We do have unpublished results that the QS signal also oscillates in L/D conditions as well. That is to be reported in another paper.

3. I am not sure whether I understand it right: Figure 5 shows that in a serpentine channel there are various numbers of hyphae (1-6). I wonder is the clock synchronization in the single hypha alone in a channel same or different from that of a single hypha present in a channel containing more than 1 hyphae?

Response: This is a very interesting question. In figure 5 we show the number of hyphae that may enter the serpentine channels, as it grows, hyphae will tend to fuse with each other, and at certain situations become one hypha or continue as individual hyphae. Hence, it may be a difficult to observe clock synchronization. We speculate that the clock synchronization will be similar since they have a chance to fuse with each other in individual serpentine channels. We could examine synchronization with channels varying with respect to the number of filaments specifically at the entrance to the serpentine channels in the beginning.

Reviewer #3 (Remarks to the Author):

The team use serpentine microfluidic devices to monitor long time growth of Neurospora crassa hyphae, along with their maintenance of circadian rhythm (monitored via a ccg-2-reporter). Although microfluidic channels have previously been used (by the same group) to study circadian rhythm in conidia, and by Hong et al. for long time imaging of individual hyphae, their full potential has remained far from realized. So the experimental work done in this paper seems appropriate.

Questions about the spatial extent of synchronized behavior are important to understanding the cell biology of this important model circadian system. Both experiments and modeling have, to date, elucidated temporal dynamics, while treating the organism as a single well-mixed compartment, and the data presented here go some way toward clarifying how nuclei and cytoplasmic cues enable different parts of the same organism to remain synchronized.

There is quite a lot of new experimental data, and I appreciated Fig 12A, especially, showing that three parts of a single hypha could maintain the same phase, despite being mm's apart. I also found the study of mean nuclear velocities along the hypha to be very interesting. I was surprised that the mean nuclear velocity in the subapical hypha is less than the speed of the growing tip, though I have some comments (below) on the importance of studying population variabilities there.

Response: We greatly appreciate your favorable comments on the work and the thoughtful comments below, which have allowed us to improve the quality of the manuscript. Thank you so much for your thoughtful critique.

There are two main sources of revision that must be made to the paper before I can recommend it for publication:

1. At the outset, the paper makes clear that synchronization within hyphae can rely on the same mechanisms as conidia, or upon intracellular sharing of signals. However, the data don't really explain the extent to which this experiment can differentiate between the two types of signalling. It seems like the model is supposed to do this work (though I don't think it does).

Response: We now include in the discussion of how one might go about distinguishing these two forms of communication.

Action: Text has been added to the discussion:

“Another limitation of the model is the effects of intrafilament vs. interfilament communication within the model. It would still be desirable to distinguish both forms of communication experimentally, and one approach to this problem is the design of new microfluidic devices to test the modes of communication. Devices are being constructed to filter for small signaling molecules³.”

I'm curious whether there are any evidences of phase differences upstream versus downstream that could demonstrate that cues are following the cytoplasmic flow. Conversely if cues are diffusing outside of the hypha, then can the investigators use their perfusion mechanism to create directional flows of cues that change the pattern of synchronization?

Response: That is a great point. We did compare the phase for different segments along the serpentine device (**Fig.12**). Interestingly, we were able to observe synchronous behavior in each of the phase trajectories. We are working directly on experimental approaches to this question which go well beyond altering perfusion by altering the microfluidics device itself. We have added a limitations section to the discussion pointing to this future work.

Action: We have added this point as a discussion item in the text:

“Another limitation of the model is the effects of intrafilament vs. interfilament communication within the model. It would still be desirable to distinguish both forms of communication experimentally, and one approach to this problem is the design of new microfluidic devices to test the modes of communication. Devices are being constructed to filter for small signaling molecules³.”

2. The mathematical modeling does not seem to me to be successful or well-considered. Reaction-advection equations are presented in the main text (I remark here that there should be a minus sign in front of the divergence of F term in equations 1 and 6), promising a relatively straightforward modification of existing well-mixed (ODE) models to include nuclear populations, nutrients and their transport. However, when we read the supplement, we discover that many extra ingredients are added for intracellular and extra-cellular transport, along with an additional signaling mechanism (with intracellular and extracellular concentrations [Si] and [Se]), couplings to nuclear division rates (which also need to be parameterized to ensure a carrying capacity, etc).

Response: The most difficult experimental challenge to the model is obtaining a “stable banding pattern” spatially as seen in race tubes on a macroscopic scale. A stable banding

pattern is to see a peak at time t for every x . This is nontrivial, and we were successful in achieving this in the model! Of course, the peaks also should have the right period as well, and they do. You are absolutely correct that the model has many additional features that can be turned on or off as the scientist chooses. These additional features come into play when the geometry of the device is changed or when the initial velocity distribution is changed with redesign of the microfluidics device or when different hypotheses about communication are used or when the epigenetic state of a gene is varied in the next generation. The sensitivity of the stable banding pattern can also be tested to intracellular or extracellular diffusion in the model with these additional features.

Richard Levins has written about three different approaches to modeling: 1) realism; 2) precision; 3) generality. Ideally, we want to maximize all three. For us, realism and precision has taken precedence over generality and heuristic approaches because to date we have been successful in this approach with the clock^{1,4}. There is substantial empirical basis for the model proposed. For example, all of the network parameters are specified as ensembles by prior published work^{1,2,4}. We have setup a path to adding further realism, such as aging septae, to identify these models with ensemble methods.

Action: We now emphasize in the text this fundamental result of a banding pattern in the results and discussion. The following text was added to the results and discussion:

“The most striking result was the stable “banding pattern” obtained in real systems, the same peak at time t for different spatial x .”

“Attaining this result constrains size of the advection zone at the beginning and ending time of the experiment. Without these cutoffs on the advection zone at the beginning and end it is difficult to achieve stable banding patterns. “

Now, even the authors of the original well-mixed models would (hopefully) admit that their models, which contain many different species, reaction-rates etc., are not well-constrained by existing experimental data, and correspondingly contain many different rates that are enormously separated in their time-scales. Here, yet more complexity is layered on, giving the investigators immense extra freedom to choose parameters and initial conditions.

Response: The complexity is needed in designing new microfluidic devices with different geometries and in accommodating different velocity distributions for the flow or the epigenetics of the system. Even though there are many parameters, that does not mean they are not methods to identify constraints on the parameters with the available data using ensemble methods to predict how the system behaves by model averaging over the ensemble. We also want to emphasize that there are features of the data that do highly constrain some of the parameters in the model. One of these features is the stable banding pattern observed (**Fig 7a**). The beginning and ending value of the advection zone size, for example, is tightly constrained. It was nontrivial to find a model with this property of a stable banding pattern. The clock network itself is also highly constrained by previous data microscopically and macroscopically^{1,4}.

Action: We have added text to the results and discussion that indicate that the stable banding pattern highly constrains the model:

“The most striking result was the stable “banding pattern” obtained in real systems, the same peak at time t for different spatial x .”

“Attaining this result constrains the size of the advection zone at the beginning and ending time of the experiment. Without these cutoffs it is difficult to achieve stable banding patterns. Besides that, the genetic network parameters are highly constrained by previous experiments^{1,4}. “

Yet, the fit to real data is not good -- importantly, in the model, there are consistent phase differences between apical, sub-apical and basal parts of the hypha, that I do not see in the real experimental data (Fig 11a).

Response: In contrast the phase differences between segments are well captured in **Fig. 12c**. The model captures quantitatively and certainly qualitatively the spatial-temporal patterning of the CCG-2 protein (**Fig. 11d**) as well as phase differences in segments (**Fig. 12c**). The parameters for the genetic network are well specified by ensemble methods using both macroscopic and microscopic experiments^{1,2,4}. We agree there are other limitations of the model, such as the phase differences subapically as septa age. Modeling is an iterative process, and we have taken the current realistic model to a nice evaluation point for the next round of experiments to test and to evaluate the models. In this modeling process we do not rely on one identified model but an ensemble of models averaged over to make predictions^{5,6}.

Action: We have added a section to the discussion discussing the evidence for and against the model. Some features are well described, but others uncover limitations of the model.

“The model presented has both some successes and limitations. The striking success is producing stable banding patterns with peaks at time x for all spatial x with the right period. This provides an explanation for the banding patterns in race tubes as well⁷. The phase curves for different segments are also well predicted by the model (**Fig. 11c**). On the other hand, there are consistent phase differences between apical, subapical and basal parts of the hyphae needing explanation. One way to accommodate these differences is to allow for septal occlusion as hyphae age⁸. These will allow additional dispersion of flow at septa. The model created is quite general and designed to accommodate different microfluidic geometries to be used in the future. It is also setup for iterative improvement by future experiments with the tools of systems biology⁴.”

Meanwhile, the 'good' agreement in Fig 11c, merely indicates that the model is correct on the period length of the circadian oscillator. Now, I want to emphasize that I do not expect models to perfectly agree with data, we learn a lot about the sufficiency of assumptions etc., when a simple model disagrees with experimental data. But this model is so complex, that its failure leaves us uncertain about which of the many assumptions are incorrect. For what it's worth, I suspect that the main problem is the lack of proper modeling of diffusion within the hypha. Diffusive terms

are included in a few of the PDE equations. But transport typically produces significant Taylor dispersion.

And in *Neurospora*, where protoplasmic flows must periodically focus down into narrow pores, small amounts of diffusion can lead to large dispersion of flows. Taylor dispersion is an odd phenomenon in that lowest diffusive elements (e.g. nuclei) can have highest dispersivities.

As I write this, I think that the best approach the authors could take is to work with a very simple, qualitative model (e.g. see the work of Peskin on single feedback loop circadian rhythm models), and to specifically test the effect of varying diffusivity within (and outside of) the cell.

Response: We agree with the reviewer that there should be additional diffusive elements around septa, and we now include text to this effect in the discussion. We also did some additional calculations of the Reynolds number in and around septal pores to examine turbulence. Assuming a viscosity similar to water inside a filament, an occluded pore size of 1 micron, and velocity within the advection zone of .1 microns/sec (outside the advection zone, the velocity is less than .01 microns/sec) in **Fig. S8**, then the Reynolds number is $10^{-7} \ll 1$, ruling out turbulence. The script for the Reynolds Number in MatLab is below:

```
function [Re] = reynolds(rho,U,L,mu)
char rho u L mu;
rho = input('Density in kg/m^3 ');
u=input ('Speed in um/s ');
L=input ('Characteristic length in um ');
mu=input ('Viscosity in Pa.s ');

u=u/1000000; %convert to m/s
L=L/1000000; %convert to m
Re=rho*u*L/mu; %units are kg/m^3, m/s, m, Pa.s
respectively

disp('Laminar flow if reynolds number <2300')
disp('Turbulent flow if reynolds number >2900')
disp('The reynolds number is ')
disp(Re)

end
```

While turbulence is ruled out, Taylor dispersion may still remain a possibility to establish further mixing around occluded pores. Hence, we carried out calculations from

experimental data (**Supplementary Table 3**), and was able to establish there is less of a possibility of Taylor dispersion affecting the flow in *Neurospora crassa*. We also included work done by Roper et al. who also agrees that Taylor dispersion is significantly small in hyphae.

Action: The following text was added:

“The model presented has both some successes and limitations. The striking success is producing stable banding patterns with peaks at time x for all spatial x with the right period. This provides an explanation for the banding patterns in race tubes as well⁷. The phase curves for different segments are also well predicted by the model (**Fig. 11c**). On the other hand, there are consistent phase differences between apical, subapical and basal parts of the hyphae needing explanation. One way to accommodate these differences is to allow for septal occlusion as hyphae age⁸. These will allow additional dispersion of flow at septae. The model created is quite general and designed to accommodate different microfluidic geometries to be used in the future. It is also setup for iterative improvement by future experiments with ensemble methods of systems biology⁴.”

“A final issue for the model is whether a combination of advection and diffusion called Taylor dispersion could lead to longer distance mixing at septal pores⁴⁴. A precursor for Taylor dispersion is that there be differences in the velocity of nuclei at the edges and those in the middle of a filament, as might be seen in the currents in a tidal river emptying into the ocean. The effect of higher velocity flow in the center of the channel has the effect of pushing a zone of higher velocity and mixing as the water leaves the river and empties into the ocean. This is an example of Taylor dispersion. The drift velocities of nuclei were measured on the edges of the filament and in the center for four hyphal tips, and there is no significant difference in velocities of nuclei dependent on their location relative to the center of the filament (**Supplementary Table S3**). This observation is consistent with the findings of others examining Taylor dispersion in *N. crassa*⁴⁵.

Additional structural issues:

The investigators should conform to best practices for data sharing, including all of their experimental data, along with the original matlab code used to solve the model (even experienced Matlab programmers will have problems understanding the printout of the code that is offered with the manuscript).

Response: The code for the model is fully documented, attached as a supplement, and a detailed manual is provided to operate the MATLAB code. All that one has to do is double click the attached MATLAB file to run the simulator. We provided a detailed manual for modifying the MATLAB file for those interested in running the simulator. All the data are provided in a Dropbox link. There is no repository large enough for the imaging data. We have indicated willingness to make data available by access to the Dropbox.

Action: The following text was added to make these points in the Materials and Methods:

Non-major considerations:

The general types of algorithms used to solve the PDEs should be described in the text; this includes the form of discretization: that they are using method of lines, what their time integration looks like, how they are treating the finite difference approximations for the derivatives to avoid numerical instability, etc.

Response: We have added a section to the materials and methods describing the solvers used. Text was added both to the modeling section entitled “The behavior of the hyphal clock model” on the numerical analysis of the model as well as to the Materials and Methods.

Action: The following text was added:

“Solving the Hyphal Clock Model. The Partial Differential Equation (PDE) system is converted into an Ordinary Differential Equation (ODE) system by a discretization of the x -coordinate on an equidistant x -grid, x_n . Specifically, if a given species of concentration $f(x, t)$ is subject to transport by diffusion inside the filament, the 2nd x -derivative of $f(x, t)$, enters into the PDE, *via* the corresponding diffusion term, and it is approximated by

$$\partial_x^2 f(x_n, t) \approx \frac{1}{(\Delta x)^2} [f(x_{n+1}, t) + f(x_{n-1}, t) - 2f(x_n, t)] .$$

Here, $\Delta x = L/N_L$ is the x -grid spacing, L is the length of the x -domain, $[0, L]$, and the x -grid points are $x_n = n\Delta x$ for $n = 0, 1, \dots, N_L$ for a chosen grid size N_L .

Similarly, if a given species of concentration $f(x, t)$ is subject to advection inside the filament, with a drift velocity field $v_f(x, t)$, the species is transported by a concentration drift or advection current, given by

$$J_f(x, t) = v_f(x, t)f(x, t).$$

The 1st x -derivative of $J_f(x, t)$ then enters into the PDE, *via* the corresponding advection term, and it is approximated by

$$\partial_x J_f(x_n, t) \approx \frac{1}{\Delta x} \left[|J_f(x_n, t)| - R \left(J_f(x_{n-1}, t) \right) - R \left(J_f(x_{n+1}, t) \right) \right]$$

where $R(J) := \max(J, 0)$ for real argument J .

The foregoing approximations to the x -derivatives preserve the particle number conservation laws that are obeyed by the respective diffusive and advective transport processes described in the underlying exact PDE system.

The resulting ODE system for the corresponding discrete dynamical variables, $f_n(t) \equiv f(x_n, t)$, is then solved by an adaptive 4th order Runge-Kutta algorithm, over a solution time interval, $[0, T]$, of duration T , with an ODE solution relative error tolerance of 10^{-6} or better. To estimate and control the numerical errors introduced by the discretization of the x -coordinate, the ODE system is solved for a doubling sequence of x -grid sizes N_L , *i.e.*, first for some initial grid size $N_L^{(0)}$, then for $N_L^{(1)} = 2N_L^{(0)}$, then for $N_L^{(2)} = 4N_L^{(0)}$ and so on. This is done until the relative change in the PDE solution values $f(x, t)$ during the (last) N_L -doubling step changes by less than some maximum allowed relative error for each species concentration f . That is, we quantify the

numerical discretization error in terms of the relative root mean square (RMS) change of $f(x, t)$ from grid size $N_L^{(v)}$ to $N_L^{(v+1)} \equiv 2N_L^{(v)}$, defined by

$$r_f^{(v)} := \frac{\text{RMS}_v[f^{(v+1)} - f^{(v)}]}{\text{RMS}_v[f^{(v)}]}.$$

Here, $f^{(v)}(x, t)$ denotes the ODE solution obtained for some species concentration $f(x, t)$ on an x -grid of size $N_L^{(v)} = 2^v N_L^{(0)}$. For any function $g(x, t)$, the quantity $\text{RMS}_v[g]$ denotes the root mean square of $g(x, t)$, over the solution time interval $[0, T]$ and over the x -grid of size $N_L^{(v)} = 2^v N_L^{(0)}$, defined by

$$\text{RMS}_v[g] := \sqrt{\frac{1}{N_L^{(v)}} \sum_{n=0}^{n=N_L^{(v)}} \frac{1}{T} \int_0^T |g(x_n^{(v)}, t)|^2 dt}.$$

The $x_n^{(v)} = (L/N_L^{(v)})n$ for $n = 0, 1, \dots, N_L^{(v)}$ are the x -grid points for grid size $N_L^{(v)}$. Note that the ODE solution values $f^{(v+1)}$, obtained for grid size $N_L^{(v+1)} = 2N_L^{(v)}$, are also defined, *i.e.*, numerically calculated, for all x -grid points $x_n^{(v)}$ of the smaller grid of size $N_L^{(v)}$, since the grid points $x_n^{(v)}$ are a subset of the grid points $x_n^{(v+1)}$. The difference function, $f^{(v+1)} - f^{(v)}$, entering into the definition of $r_f^{(v)}$, is therefore also defined for all x -grid points $x_n^{(v)}$.

The t -integrals in the foregoing definition of $\text{RMS}_v[f]$ are approximated by applying the trapezoidal numerical integration rule, with summation over an equidistant ODE solution output t -grid of grid spacing Δt . For the numerical results presented here, we used $\Delta t = 1$ h for solution time durations of up to $T = 240$ h.

If we start from a sufficiently large initial x -grid size, $N_L^{(0)}$, the doubling sequence can be terminated already after one doubling step, *i.e.*, at $v = 0$."

The authors note that there are three main mechanisms by which conidia can maintain synchronous oscillations: contact and diffusible QS and stochastic synchronization. Two papers reference stochastic synchronization, while Stan Liebler's paper seems to be about robustness against stochastic effects and seems less well located. The text says three additional mechanisms exist for hyphae, but I think only intracellular synchronization is actually distinctive to syncytia, cf. single fungal cells.

Response: Your point about Liebler's paper is well taken. We have removed this reference. There are two additional mechanisms reported in the literature. The first is gating by the cell cycle. It is reported as a novel mechanism in and of itself by PNAS⁹. The second mechanism is having a growth wave with local mixing at the wave front. This seems different from a QS mechanism within a filament.

Replicate numbers are not made clear or uncertainties considered: is the important figure 11a based on a single hypha? How do other hyphae compare? How do the

period uncertainties that come from the very noisy data (see Fig 6) affect the results?

Response: The data are in **Fig. 11d**. We now indicate in the legend the number of hyphae used. In that there is a possibility you mean **Fig. 12a**, we now clarify the number of channels used for computing **Fig. 12**. In the same spirit we have also taken **Fig. 9** and made clear the number of hyphae involved in periodogram calculations. Thank you for pointing out the need for clarification.

Actions: Legends to **Fig. 9, 11, and 12** have been clarified on the number of hyphae involved.

** See the Nature Portfolio author and referees' website at www.nature.com/authors for information about policies, services and author benefits

Communications Biology is committed to improving transparency in authorship. As part of our efforts in this direction, we are now requesting that all authors identified as 'corresponding author' create and link their Open Researcher and Contributor Identifier (ORCID) with their account on the Manuscript Tracking System prior to acceptance. ORCID helps the scientific community achieve unambiguous attribution of all scholarly contributions. You can create and link your ORCID from the home page of the Manuscript Tracking System by clicking on 'Modify my Springer Nature account' and following the instructions in the link below. Please also inform all co-authors that they can add their ORCIDs to their accounts and that they must do so prior to acceptance.

If you experience problems in linking your ORCID, please contact the Platform Support Helpdesk.

This email has been sent through the Springer Nature Tracking System NY-610A-NPG&MTS

Confidentiality Statement:

This e-mail is confidential and subject to copyright. Any unauthorised use or disclosure of its contents is prohibited. If you have received this email in error please notify our Manuscript Tracking System Helpdesk team at <http://platformsupport.nature.com>.

Details of the confidentiality and pre-publicity policy may be found here <http://www.nature.com/authors/policies/confidentiality.html>

Privacy Policy | Update Profile

- 1 Cheong, J. H. *et al.* The macroscopic limit to synchronization of cellular clocks in single cells of *Neurospora crassa*. *Scientific Reports* **12**, 6750 (2022).
- 2 Deng, Z. *et al.* Single cells of *Neurospora crassa* show circadian oscillations as well light entrainment and temperature compensation. *IEEE Access* **7**, 49403-49417 (2019).
- 3 Mossu, A. *et al.* A silicon nanomembrane platform for the visualization of immune cell trafficking across the human blood–brain barrier under flow. *Journal of Cerebral Blood Flow & Metabolism* **39**, 395-410 (2019).
- 4 Dong, W. *et al.* Systems biology of the clock in *Neurospora crassa*. *PLoS one* **3**, e3105 (2008).
- 5 Yu, Y. *et al.* A genetic network for the clock of *Neurospora crassa*. *Proceedings of the National Academy of Sciences of the United States of America* **104**, 2809-2814 (2007).
- 6 Battogtokh, D., Asch, D. K., Case, M. E., Arnold, J. & Schuttler, H. B. An ensemble method for identifying regulatory circuits with special reference to the *qa* gene cluster of *Neurospora crassa*. *P Natl Acad Sci USA* **99**, 16904-16909 (2002).
- 7 Gooch, V. D. *et al.* Fully codon-optimized luciferase uncovers novel temperature characteristics of the *Neurospora* clock. *Eukaryotic cell* **7**, 28-37 (2008).
- 8 Markham, P. Occlusions of septal pores in filamentous fungi. *Mycological Research* **98**, 1089-1106 (1994).
- 9 Paijmans, J., Bosman, M., Wolde, P. R. t. & Lubensky, D. K. Discrete gene replication events drive coupling between the cell cycle and circadian clocks. *PNAS USA* **113**, 4063-4068 (2015).

Reviewers' comments:

Reviewer #3 (Remarks to the Author):

I like reading this story. The revised manuscript addressed most of my comments given in last round.

For the response to my third question, regarding the synchronization with channels varying with respect to the number of filaments, I suggest the authors include the discussion on this aspect and the possible (planned) experiments in future to resolve this issue in the manuscript (either in the section "Variation of hyphal number growing along serpentine channels" or in "Discussion").

Reviewer #5 (Remarks to the Author):

A previous reviewer had commented that the mathematical model used in this work seems very cumbersome and is probably unnecessarily complex. I agree with that comment, especially since the authors do not justify why they make the particular modelling choices that they describe, and how they arrived at the model presented here. The documentation of the Matlab code contains a lot of partial differential equations which are provided without justification or description, as well as a lot of new parameters whose settings are completely unknown. On the other hand, the ordinary differential equations for the circadian clock model (main text) are not even presented anywhere. Overall, the description and explanation of the model are highly incomplete, and I found it very difficult to maintain an overview of what exactly is modelled here.

In response to that reviewer comment, the authors responded that the model reproduces a “stable banding pattern” that is observed in the actual system. However, this feature alone does not validate the model. If achieving such a feature with the given model is challenging, then one has to ask why this is the case, since this feature seems to be routinely observed in experiments. In my view, this fact implies that the modelled biological system has a built-in robustness that the model cannot capture, for example because the model is wrong or incomplete.

Then there is the issue of model parameterization and the fact that the described model contains a very large number of unobservable species and parameters. The authors respond that the model is highly constrained by previous data, but 1) do not explain what these data are, and 2) which parameters are constrained. They also do not present any study on the sensitivity of the model to the unknown parameters.

Overall, I fail to see what purpose that modelling serves in this work. The text “oscillates” between (interesting) experimental findings and modelling, without a clear sense of direction. The authors admit that the processes that drive synchronization over long spatial scales are not exactly known, but modelling is not used to clarify this point either. On the other hand, it is unclear which aspects of the model need to be improved or changed, as the authors claim that the model is already precise enough (which I do not see).

Response to Reviewers

Thank you for the opportunity to prepare a revision. We took full advantage of the comments of each reviewer. Below is a point-by-point response. The reviewer comments are in bold followed by our Response and Action to accommodate the reviewer comment. I have uploaded both a clean copy and copy with all revisions tracked.

Reviewers' comments:

Reviewer #3 (Remarks to the Author):

I like reading this story. The revised manuscript addressed most of my comments given in last round.

Response: Thank you for all your comments on improving the manuscript. We are deeply appreciative.

For the response to my third question, regarding the synchronization with channels varying with respect to the number of filaments, I suggest the authors include the discussion on this aspect and the possible (planned) experiments in future to resolve this issue in the manuscript (either in the section "Variation of hyphal number growing along serpentine channels" or in "Discussion").

Response: We have followed the reviewer's suggestion and added text to the discussion on this point.

Action: The following text was added to the discussion:

“Another question is whether or not synchronization varied with the number of filaments within a channel. One way to address this question is to design a new device that allows two adjacent channels of varying width to assay filament synchronization. Filaments in adjacent channels could be assayed for synchronization, while varying the width of a channel could be used to control the number of filaments.”

Reviewer #5 (Remarks to the Author):

A previous reviewer had commented that the mathematical model used in this work seems very cumbersome and is probably unnecessarily complex. I agree with that comment, especially since the authors do not justify why they make the

particular modelling choices that they describe, and how they arrived at the model presented here.

Response: We have developed a model supplement developing and describing the model in detail. We also tabulate the parameters used from previous ensemble runs used in this paper. Finally, at each step, we give the rationale for assumptions made by the model. The complexity of the model reflects what we know about the clock in *N. crassa* and gives us flexibility in clock studies in a variety of microfluidic devices. We appreciate the reviewer's interest in seeing a more detailed model description and have written full description of the model in the model supplement.

Action: We have added a model supplement that fully justifies the choices made in the model and how we arrived at the parameter specifications.

The documentation of the Matlab code contains a lot of partial differential equations which are provided without justification or description, as well as a lot of new parameters whose settings are completely unknown.

Response: We thank the reviewer for the requested clarification and for helping to build a more substantial publication. We have rewritten the model section to highlight the main features of the model for greater clarity and aligned it with the new Model Supplement with a more detailed description. We now present a detailed development of the model in a Model Supplement, including a precise description of the partial differential equations. We tabulated and extracted all of the parameters used from the MATLAB code in the Model Supplement and Supplementary Materials to make it easier for the reader. We also indicate how those parameters are specified empirically from ensemble runs in previous citations. For example, all the rate constants are taken from prior publications using ensemble methods at the single-cell level and macroscopic scale.

Action: We have justified the model in the new model supplement, with tabulated parameters used. Also, the following text has been added to the main body in the model section:

“It is described in detail in the Model Supplement, including the complete model parameter set and the numerical PDE solution method which we have employed to generate the model results shown in Fig.11 below. The implementation of the model is also described in a manual for the MATLAB code we have used to compute the PDE solution.”

On the other hand, the ordinary differential equations for the circadian clock model (main text) are not even presented anywhere. Overall, the description and

explanation of the model are highly incomplete, and I found it very difficult to maintain an overview of what exactly is modelled here.

Response: We thank you for the suggestion to make the paper more self-contained. In the model supplement, we have tabulated the rate constants specifying the ordinary differential equations for the circadian clock. We have also presented all of the ordinary differential equations in the Model Supplement derivable from Figure 4 in the main text.

Action: Please see Model Supplement.

In response to that reviewer comment, the authors responded that the model reproduces a “stable banding pattern” that is observed in the actual system. However, this feature alone does not validate the model. If achieving such a feature with the given model is challenging, then one has to ask why this is the case, since this feature seems to be routinely observed in experiments. In my view, this fact implies that the modelled biological system has a built-in robustness that the model cannot capture, for example because the model is wrong or incomplete.

Response: We agree with the reviewer that it is essential to ask why achieving a stable banding pattern was so tricky. We could plead a smarter person would have immediately seen the solution. We did not. Once we found the solution, we did describe the solution in the discussion. The solution was simple – constrain the size of the advection zone at the experiment’s beginning and end; however, this model feature does not produce a brittle model. It is simply a feature of the model we failed to see that was needed. It also happens to be a feature that we measured. With this feature, an extensive collection of models produces stable banding patterns, such as those with rate coefficients in the ensemble. We include a second model with different parameters to demonstrate this in the Model Supplement and the Supplementary Materials.

Action: We have added the following clarifying sentence in the discussion that the models we found are not brittle:

“The cutoffs, however, do not constrain the ensemble of rate coefficients for the clock network – they are a feature of the physical process of advection.”

Then there is the issue of model parameterization and the fact that the described model contains a very large number of unobservable species and parameters. The authors respond that the model is highly constrained by previous data, but 1) do not explain what these data are, and 2) which parameters are constrained. They also do not present any study on the sensitivity of the model to the unknown parameters.

Response: Thank you for the clarification request. In the model supplement, we describe how we obtain the parameters and include them in the tables in the Model Supplement using ensemble methods in prior published work. These ensemble methods were developed to handle models with many parameters and limited data (PNAS 104: 2809). Furthermore, we indicate that the ensemble method provides an automatic sensitivity analysis through the ensemble as a byproduct. We also indicate the constraint on the cutoffs for the advection zone. We also indicate the ensemble fitting providing 40,000 models that yield kinetic parameters consistent with available microfluidic and macroscopic data previously published. We cite the published work that produced the model ensembles used here.

Action: We provide the citations to published work for parameter values in the new Model Supplement and method for doing a sensitivity analysis, namely through the ensemble methods. We also provide the parameters in the model supplement as well.

Overall, I fail to see what purpose that modelling serves in this work. The text “oscillates” between (interesting) experimental findings and modelling, without a clear sense of direction. The authors admit that the processes that drive synchronization over long spatial scales are not exactly known, but modelling is not used to clarify this point either. On the other hand, it is unclear which aspects of the model need to be improved or changed, as the authors claim that the model is already precise enough (which I do not see).

Response: Again, we are grateful to the reviewer for the comments to make this a more substantial publication and the opportunity to clarify the role of the model here. We use long-distance synchronization as a phenomenon to give a unifying flow to the text on modeling. At the beginning of the model, we motivate our model, as precisely the framework for examining long-distance synchronization. We hypothesize that the advection wave is a model feature that explains the phase synchronization measured and give graphical presentations of this wave in Figure 11. Our experimental results in Table 2 also provide evidence for long distance synchronization across channels. Firstly, we show that the stable banding patterns of the data and model resemble each other. Further model validation is done by measuring phase synchronization through the Hilbert Phase in different segments of one channel as shown in Figure 12 and comparing same to model predictions. To better highlight the thread of this argument, we have modified the text below to highlight how the model produces synchronization of segments through the passing advection wave.

Another example of how the model informed the measurements in this paper was the concept of an advection zone (revised Figure 3). We would not have introduced the concept of an advection zone without the model. This concept allows us to frame biological hypotheses about how nuclei move in filaments. See the text below added.

Action: We have added the following clarifying text as a unifying thread for the paper:

“Are there new ways to conceptualize how synchronization of clock filaments arises over long distances? Again, the dynamics of an advancing wave of cytoplasm within a growing filament may provide an answer about long-distance synchronization that could not be envisioned without the model. These are the reasons that a model must be considered for a growing filament.”

We pointed out the evidence for long-distance synchronization in the section on
“Synchronization of hyphae in serpentine channels”

“at long distances, one of the intriguing aspects of filament synchronization.”

We display the advection wave in Figure 11 and highlight its importance as a possible mechanism for phase synchronization:

“This wave front provides a new mechanism for long-distance synchronization (Table 2), in which advection and diffusion cause local mixing, but long-distance synchronization is achieved by carrying clock mRNAs and proteins in the advection wave.”

We provide additional Hilbert phase data between segments to bolster the case for long-distance synchronization in Figure 12:

“consistent with the advection wave producing phase synchronization”.

We conclude with existing text in the discussion on the advection wave as a cause of long distance synchronization.

With regard to the advection zone concept derived from the model, we added the following text:

“These features to be measured are motivated by the model.”

“This concept arose directly from the consideration of the hyphal clock model.”

Thank you again for the opportunity to prepare a revision and a stronger manuscript. We are grateful for all the comments by the reviewers. We have used all their comments to make a better manuscript.

Best wishes,

Jonathan Arnold

REVIEWERS' COMMENTS:

Reviewer #5 (Remarks to the Author):

I would like to thank the authors for their detailed reply to my comments. I feel that all my main concerns have been addressed, and I do not have any further comments to make.